# Myosin II isoforms play distinct roles in *adherens* junction biogenesis

**Mélina L Heuzé[1†]\*, Gautham Hari Narayana Sankara Narayana[1†], Joseph D'Alessandro[1], Victor Cellerin[1], Tien Dang[1], David S Williams[2], Jan CM Van Hest[3], Philippe Marcq[4], René-Marc Mège[1]\*, Benoit Ladoux[1]\***

[1]Institut Jacques Monod, Université de Paris and CNRS UMR 7592, Paris, France; [2]Department of Chemistry, College of Science, Swansea University, Swansea, United Kingdom; [3]Institute for Complex Molecular Systems, Eindhoven University of Technology, Eindhoven, Netherlands; [4]Laboratoire Physique et Mécanique des Milieux Hétérogènes, Sorbonne Université and CNRS UMR 7636, Paris, France

**Abstract** Adherens junction (AJ) assembly under force is essential for many biological processes like epithelial monolayer bending, collective cell migration, cell extrusion and wound healing. The acto-myosin cytoskeleton acts as a major force-generator during the de novo formation and remodeling of AJ. Here, we investigated the role of non-muscle myosin II isoforms (NMIIA and NMIIB) in epithelial junction assembly. NMIIA and NMIIB differentially regulate biogenesis of AJ through association with distinct actin networks. Analysis of junction dynamics, actin organization, and mechanical forces of control and knockdown cells for myosins revealed that NMIIA provides the mechanical tugging force necessary for cell-cell junction reinforcement and maintenance. NMIIB is involved in E-cadherin clustering, maintenance of a branched actin layer connecting E-cadherin complexes and perijunctional actin fibres leading to the building-up of anisotropic stress. These data reveal unanticipated complementary functions of NMIIA and NMIIB in the biogenesis and integrity of AJ.

DOI: https://doi.org/10.7554/eLife.46599.001

**\*For correspondence:**
melina.heuze@ijm.fr (MLH);
rene-marc.mege@ijm.fr (RMM);
benoit.ladoux@ijm.fr (BL)

[†]These authors contributed equally to this work

**Competing interests:** The authors declare that no competing interests exist.

## Introduction

Tissue integrity and plasticity rely on cell-cell adhesion and cell contractility. The formation, remodeling and disassembly of cell-cell adhesions are fundamental events accompanying all stages of morphogenesis, tissue homeostasis and healing. AJ mediated by E-cadherin/catenin complexes are key elements of epithelial cell-cell adhesions and the first ones to assemble upon contact initiation (*Adams et al., 1998*; *Green et al., 2010*; *Takeichi, 2014*). They provide strong mechanical coupling between neighboring cells through association with the acto-myosin cytoskeleton (*Mège and Ishiyama, 2017*).

The assembly of de novo AJ is crucial for cell-cell rearrangement (*Cavey et al., 2008*; *Maître et al., 2015*), tissue closure (*Jacinto et al., 2002*) and the maintenance of epithelial cell integrity during wound healing or cell extrusion (*Harris et al., 2014*; *Kocgozlu et al., 2016*; *Wood et al., 2002*). During de novo cell-cell contact formation, initial contact between facing lamellipodia induces immediate clustering of cadherin molecules by trans- and cis-oligomerization (*Adams et al., 1998*; *Yap et al., 1997*; *Strale et al., 2015*; *Mège et al., 2006*). Subsequent signaling events involving Rho GTPases trigger local remodeling of the actin cytoskeleton through Arp2/3- or formin-mediated actin polarization in the vicinity of AJs (*Grikscheit et al., 2015*; *Kovacs et al., 2002*; *Yamada and Nelson, 2007*). These cytoskeletal rearrangements drive the expansion of cell-cell contacts and inter-cellular adhesion strengthening (*Green et al., 2010*; *Krendel and Bonder, 1999*; *Chu et al., 2004*).

Non-muscle Myosin II (NMII) has emerged as a fundamental player in force-generation and force-transmission at AJ both in vitro and in vivo (*Borghi et al., 2012*; *Curran et al., 2017*; *Ladoux and Mège, 2017*). NMII is essential for epithelial tissue architecture (*Salomon et al., 2017*), epithelial tissue morphogenesis (*Munjal and Lecuit, 2014*), tissue repair (*Tamada et al., 2007*; *Begnaud et al., 2016*) and cell extrusion (*Rosenblatt et al., 2001*). NMII protects junctions from disassembly during development (*Weng and Wieschaus, 2016*) and provides the mechanical tugging force necessary for AJ reinforcement (*Liu et al., 2010*). In endothelial cells, NMII is recruited early in filopodia-mediated bridge bundles and its activity is required for accumulation of VE-cadherin in nascent AJs (*Hoelzle and Svitkina, 2012*). In epithelial cells, NMII favors local concentration of E-cadherin at cell-cell contacts (*Shewan et al., 2005*; *Smutny et al., 2010*) and it is enriched at the edges of elongating junctions where it drives contact expansion in response to RhoA (*Yamada and Nelson, 2007*; *Krendel and Bonder, 1999*).

In mammalian cells, NMII heavy chains exist as three different isoforms: NMIIA, NMIIB and NMIIC encoded by *MYH9*, *MYH10* and *MYH14* genes, respectively (*Conti and Adelstein, 2008*; *Vicente-Manzanares et al., 2009*). NMIIA and NMIIB are widely expressed whereas NMIIC is not detected in several tissues (*Ma et al., 2010*). Despite structural similarities, NMIIA and NMIIB isoforms have been assigned both redundant and specific functions depending on cell types and processes (*Beach and Hammer, 2015*). NMIIA and NMIIB exhibit different ATPase activities and actin-binding properties (*Wang et al., 2003*; *Kovács et al., 2003*; *Kovács et al., 2007*; *Billington et al., 2013*), in addition to their specific C-terminal tails that confer them unique functions (*Sandquist and Means, 2008*; *Juanes-Garcia et al., 2015*; *Chang and Kumar, 2015*). These two isoforms can exist as activated monomers in cells, but they can also co-assemble as homotypic and heterotypic filaments (*Shutova et al., 2014*; *Beach et al., 2014*). NMIIA and NMIIB play both unique and overlapping roles in vivo (*Skoglund et al., 2008*; *Wang et al., 2011*; *Haque et al., 2017*; *Ridge et al., 2017*; *Conti et al., 2004*; *Tullio et al., 1997*). In cells migrating on 2D surfaces, NMIIA localizes at the cell front, limits lamellipodial protrusive activity and reduces 2D cell migration speed by regulating focal adhesions dynamics and traction forces (*Doyle et al., 2012*; *Betapudi et al., 2006*; *Cai et al., 2006*; *Jorrisch et al., 2013*). NMIIB localizes at the cell rear and is required for front-back polarity and tail retraction (*Betapudi et al., 2006*; *Cai et al., 2006*; *Jorrisch et al., 2013*; *Kolega, 2003*; *Sandquist et al., 2006*; *Vicente-Manzanares et al., 2008*; *Vicente-Manzanares et al., 2011*; *Betapudi, 2010*; *Shutova et al., 2017*). In 3D, NMIIA favors cell displacement (*Doyle et al., 2012*; *Betapudi et al., 2006*; *Cai et al., 2006*; *Jorrisch et al., 2013*; *Shih and Yamada, 2010*) while NMIIB drives nuclear translocation (*Thomas et al., 2015*). NMIIB also plays a determinant role in durotaxis (*Raab et al., 2012*).

While the roles of NMII isoforms in cell motility on ECM have been extensively studied, very little is known on their respective functions in AJs organization. Yap and collaborators have reported that NMIIA and NMIIB both localize at apical junction complexes of polarized MCF-7 cells (*Smutny et al., 2010*; *Gomez et al., 2015*). Upon specific isoform expression silencing, they further proposed that NMIIA may favor the accumulation of E-cadherin in the AJ belt while NMIIB may stabilize the associated perijunctional actin ring, reinforce junctions and prevent them from disruptive forces (*Smutny et al., 2010*). Ozawa reported using CRISPR-Cas9 gene invalidation that NMIIA was required to assemble junctional complexes (*Ozawa, 2018*). Svitkina and collaborators reported an association of NMIIA with contractile actin bundle running parallel to linear AJ in endothelial cells, but failed to precisely localize NMIIB (*Efimova and Svitkina, 2018*). Here we further explore the functions of NMII isoforms in epithelial AJ biogenesis using an in vitro system based on chemically-switchable micro-patterns, whereby we can control the time and location of a new contact forming between two single cells on a matrix-coated surface.

## Results

### In vitro system for the study of early cell-cell contacts

In order to study early AJ biogenesis, pairs of GFP-E-cadherin expressing MDCK cells were plated on arrays of 5 µm-distant fibronectin-coated micro-patterns surrounded by switchable cytorepulsive surfaces (*van Dongen et al., 2013*). After complete spreading, the confinement imposed by the micro-patterns was released by addition of an RGD-motif containing modified peptide that switched

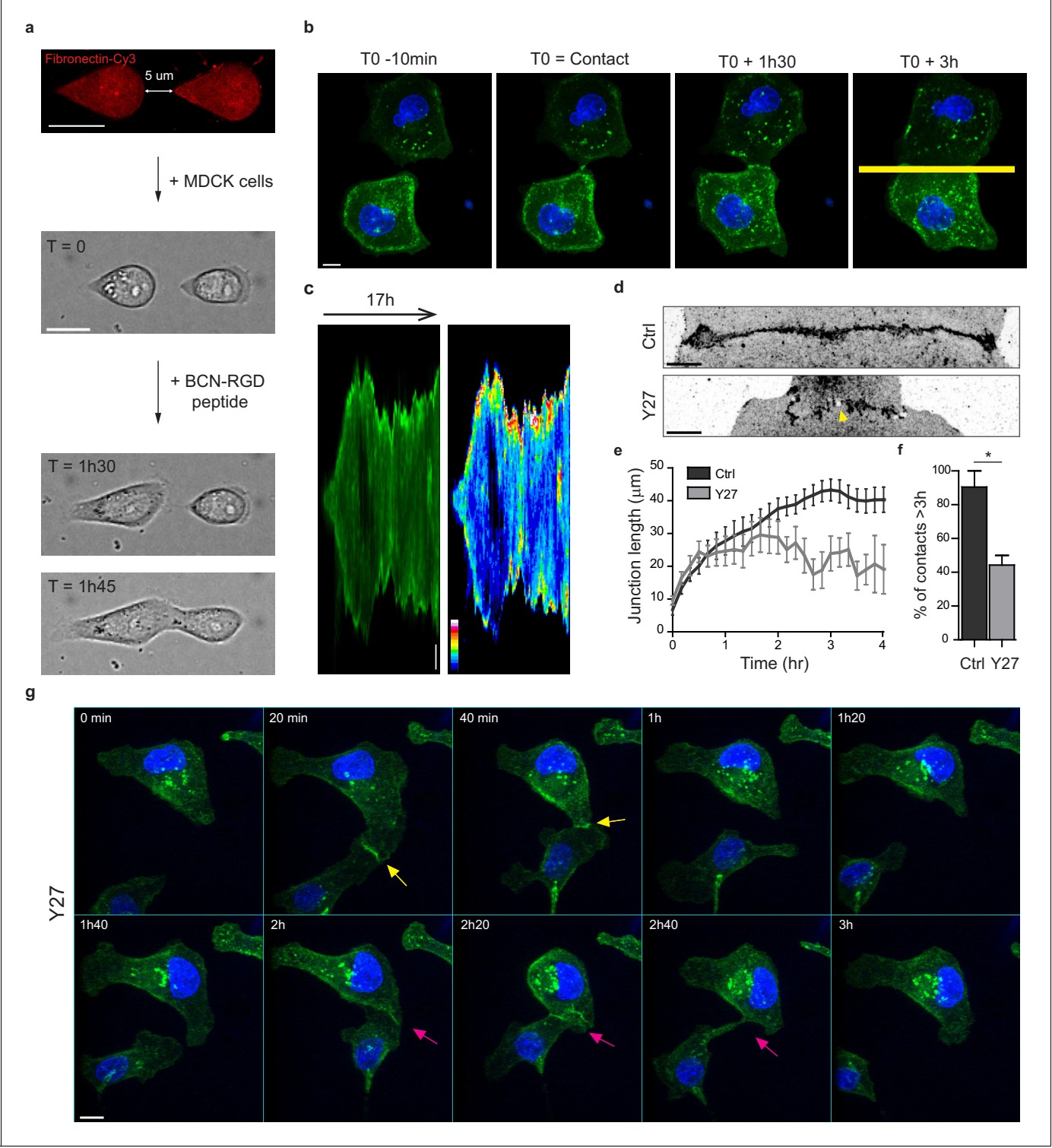

**Figure 1.** Development of an in vitro system for the study of junction biogenesis. (a) Sequential steps for controlled initiation and visualization of junction biogenesis. The two cells are initially confined on a pair of fibronectin-coated 5 µm-away patterns ($T_0$). When desired, the cell confinement is released by addition of BCN-RGD peptide, inducing cell spreading and kissing within a few hours. Scale Bar: 10 µm. (b) Spinning disk image sequence showing contact extension between two MDCK cells expressing GFP-E-cadherin and stained with Hoechst. Scale bar: 10 µm. (c) Kymographs of the junction forming in panel b, generated from the yellow line, shown in green and in pseudocolor to highlight GFP-E-cadherin accumulation at junction tips. The junction axis was realigned horizontally for some time points in order to generate the kymograph on a long time scale. Scale bar: 5 µm. (d) Representative confocal images of β-catenin-stained junctions from MDCK cell doublets. The arrow points at small holes frequently observed within

*Figure 1 continued on next page*

*Figure 1 continued*

Y27-treated junctions. The cells were fixed 20 hr after addition of BCN-RGD alone or BCN-RGD + Y27 (50 µM). Scale bar: 10 µm. (e) Graphs showing the evolution of junction length in function of time after contact initiation in Ctrl and Y27-treated MDCK cell doublets. Y27 (50 µM) was added with BCN-RGD. Data are represented as mean + /- SEM. n = 13 and 12 cell doublets from two and three independent experiments, respectively. (f) Bar graph of the percentage of cell doublets that stay in contact for more than 3 hr in Ctrl and Y27-treated MDCK cells, respectively. Data are represented as mean + /- SEM. n = 13 and 12 cell doublets from two and three independent experiments, respectively. Bonferroni statistical tests were applied for p value. (g) Spinning disk image sequence of GFP-E-cadherin-expressing MDCK cells pre-stained with Hoechst in the presence of Y27 (50 µM). The sequence starts 3 hr after addition of BCN-RGD + Y27. The arrows highlight transient contacts forming under these conditions. Scale bar: 10 µm.

DOI: https://doi.org/10.7554/eLife.46599.002

The following source data and figure supplements are available for figure 1:

**Source data 1.** Development of anin vitrosystem for the study of junction biogenesis.
DOI: https://doi.org/10.7554/eLife.46599.005
**Figure supplement 1.** Reversal of nucleus-centrosome polarity axis after cell-cell contact.
DOI: https://doi.org/10.7554/eLife.46599.003
**Figure supplement 1—source data 1.** Reversal of nucleus-centrosome polarity axis after cell-cell contact.
DOI: https://doi.org/10.7554/eLife.46599.004

the surface surrounding patterns from a cytorepulsive to an adhesive surface (*Figure 1a* and *Figure 1—figure supplement 1a*). Junction biogenesis was monitored by confocal spinning disk microscopy (*Figure 1b*, *Video 1*). Within 2 hr, cells extended lamellipodia in random directions and approximately 50% of the pairs of cells contacted within 12 hr. The junction extended reaching a plateau at 40–45 µm length in around 3 hr (*Figure 1c,e*). As previously described (*Yamada and Nelson, 2007*), GFP-E-cadherin accumulated at the edges of the junction (*Figure 1c*). Once reaching this maximal length, the junction was maintained while showing dynamic retraction-elongation events (*Figure 1c*). Importantly, in 98 + /- 2% of the cases, cell-cell contacts were stable and lasted above 3 hr and up to 22 hr (*Figure 1c,f*). Analysis of the nucleus-centrosome axis relative to the junction axis showed a relocalization of the centrosome towards the lamellipodia opposite to the cell-cell

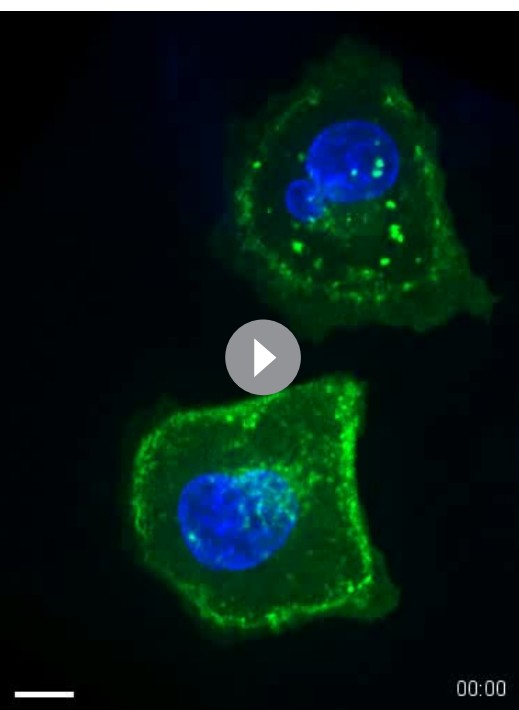

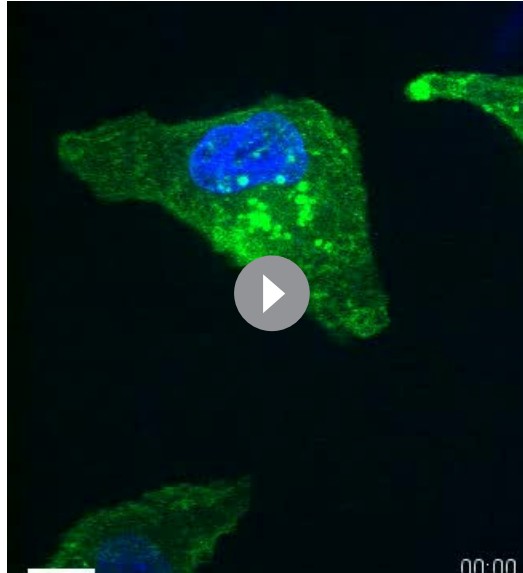

**Video 1.** Dynamic of junction formation on reversible micropatterns. Spinning disk movie showing contact formation between two MDCK cells expressing GFP-E-cadherin and stained with Hoechst. Scale bar: 10 µm.
DOI: https://doi.org/10.7554/eLife.46599.006

**Video 2.** Dynamic of junction formation in Y27-treated cells. Spinning disk movie of MDCK cells expressing GFP-E-cadherin, stained with Hoechst and treated with 50 µM Y27. Scale bar: 10 µm.
DOI: https://doi.org/10.7554/eLife.46599.007

contact within one hour (*Figure 1—figure supplement 1b–d*), as previously reported in different systems and cell types (*Desai et al., 2009*; *Dupin et al., 2009*; *Burute et al., 2017*; *Rodríguez-Fraticelli et al., 2012*). However, although MDCK cells antipolarized in the doublet as if they were initiating a contact inhibition of locomotion, they remained attached to each other in contrast to more mesenchymal cells that proceed with cell separation following repolarization (*Stramer and Mayor, 2017*). Together, these observations show that this in vitro model system is suitable for the study of early cell-cell contacts at high spatial-temporal resolution.

## NMIIA and NMIIB orchestrate junction biogenesis

To evaluate the involvement of NMII-generated actomyosin contractility in junction biogenesis, we monitored junction formation in cells treated with the ROCK inhibitor Y27632 (*Figure 1g* and *Video 2*). Y27-treated cells exhibited irregular junctions with small digitations and empty spaces and did not elongate as much as control cells (*Figure 1d,e*). They were strongly affected in their capacity to maintain cell-cell contacts, half of the doublets separating before 3 hr (*Figure 1f,g* and *Video 2*). Similar results were observed after treating cells with the NMII ATPase activity inhibitor blebbistatin (data not shown) indicating that NMII activity is required for proper junction elongation and stabilization. Furthermore, NMII was required for the centrosome repolarization, as we could not observe any preferential orientation of the nucleus-centrosome axis in Y27-treated doublets (*Figure 1—figure supplement 1d*).

Next, we explored the involvement of the two NMII isoforms in junction biogenesis. NMIIA has been reported to be by large the major isoform of NMII expressed in MDCK cells (*Ma et al., 2010*). However, immunostainings revealed that the three isoforms, NMIIA, NMIIB and NMIIC could be detected in MDCK cells. NMIIA and NMIIC fully co-localized to similar structures, which was not the case for NMIIB (*Figure 3—figure supplement 1a,b*). For these reasons, we decided to focus on NMIIA and NMIIB isoforms. Expression of each isoform was silenced in GFP-E-cadherin MDCK cells by stable transfection of specific ShRNA encoding plasmids, leading to an inhibition of expression of around 60–70% (*Figure 2a,b* and *Figure 2—figure supplement 1a,b*). The analysis of cell-cell contact formation in cell doublets by live-imaging (*Video 3*) revealed that NMIIB knock-down (NMIIB KD) cells formed and extended intercellular junctions very similar to control (Ctrl) cells (*Figure 2c–f*). In contrast, almost half of NMIIA knock-down (NMIIA KD) cell doublets were unable to sustain contacts more than 3 hr, and when they did so, these contacts remained shorter than for Ctrl or NMIIB KD cell doublets (*Figure 2c–f*), similar to what was observed in Y27-treated cell doublets. NMIIB KD doublets, despite their ability to maintain cell-cell contacts for longer times, formed twisted junctions that were significantly less straight than Ctrl and NMIIA KD cells and deviated significantly more from their initial orientation (*Figure 2g,h*). These defects in NMIIB KD cells were already observed at early stages of junction biogenesis and were associated to the formation of large extensions of junctional membrane (*Figure 2i*, arrows). Together, these results show that both NMIIA and NMIIB are required for the biogenesis of stable AJs, albeit with different contributions; NMIIA favors temporal stability whereas NMIIB ensures the straightness and spatial stability of the junctions, which is in agreement with different contributions of NMIIA and NMIIB in mature junctions (*Smutny et al., 2010*).

## NMIIB preferentially localizes to a junctional actin pool distinct from perijunctional NMIIA-associated contractile fibres

To better understand the respective roles of NMIIA and NMIIB in junction biogenesis, we next studied their subcellular localization at nascent cell-cell contacts in cell doublets. Immunostainings revealed a differential localization of the two isoforms relative to the junction. Anti-NMIIA antibodies stained actin bundles that were parallel to the junction, setting at 1 to 2 μm from it, but did not stain the junctional area. NMIIA was also found associated to actin cables parallel to the cortex of non-junctional membranes (*Figure 3a,c* and *Figure 3—figure supplement 1a*) in addition its association to the classical ventral stress fibres. NMIIB immunostaining was also present on some perijunctional actin bundles but, in contrast with NMIIA, was strongly associated with the junctional plasma membranes as well as with a cytoplasmic network (*Figure 3b,d* and *Figure 3—figure supplement 1b*), that was identified as the vimentin intermediate filament network as reported by Menko and

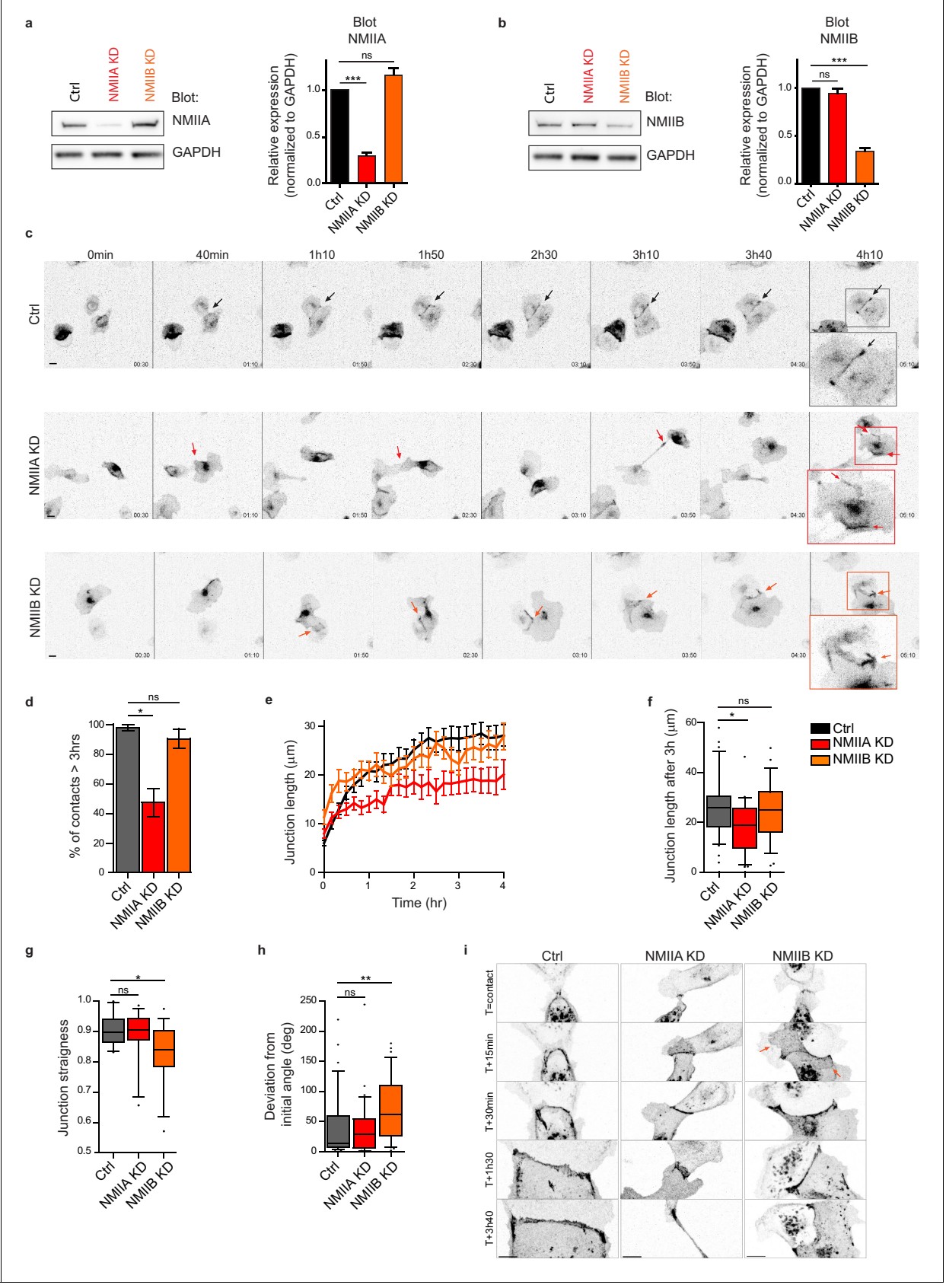

**Figure 2.** NMIIA and NMIIB are both required for proper junction biogenesis. (a, b) Left panels: Representative immunoblots showing the isoform specific knockdown of NMIIA (a) and NMIIB (b) in NMIIA KD and NMIIB KD MDCK cells. GAPDH expression levels were used as loading controls. Right panels: Bar graphs showing the relative expression level of NMIIA and NMIIB proteins in Ctrl, NMIIA KD and NMIIB KD cells normalized to GAPDH expression levels. Data are represented as mean + /- SEM from three independent experiments. Kruskall-Wallis statistical tests were applied for p value. (c) Representative epifluorescence image sequences of GFP-E-cadherin over a time course of 5 hr showing the dynamics of junction formation at low magnification in Ctrl, NMIIA KD and NMIIB KD MDCK cells. The arrows indicate the position and the orientation of the junctions. Scale bar: 10 μm. (d) Bar graph of the percentage of cell doublets that stay in contact for more than 3 hr. Data are represented as mean + /- SEM. Tukey's multiple comparison statistical tests were applied for p value. n = 36, 37 and 31 cell doublets for Ctrl, NMIIA KD and NMIIB KD cells respectively, from three independent experiments. (e) Plots showing the evolution of junction length in function of time for Ctrl, NMIIA KD and NMIIB KD cell doublets. Data are represented as mean + /- SEM. n = 40, 43 and 35 cell doublets for Ctrl, NMIIA KD and NMIIB KD cells respectively, from four independent experiments. (f) Box and whiskers graphs representing the junction length after 3 hr after contact, for Ctrl, NMIIA KD and NMIIB KD cell doublets. n = 34, 21 and 28 cell doublets for Ctrl, NMIIA KD and NMIIB KD cells respectively, from four independent experiments. (g) Box and whiskers graphs showing the junction straightness (calculated as the euclidean/accumulated length ratio) in Ctrl, NMIIA KD and NMIIB KD cell doublets 2 hr after contact. n = 12, 15 and 17 cell doublets for Ctrl, NMIIA KD and NMIIB KD cells respectively, from three independent experiments. (h) Box and whiskers graph showing the angular deviation of junctions during the three first hours of contact in Ctrl, NMIIA KD and NMIIB KD cell doublets. n = 35, 30 and 32 cell doublets for Ctrl, NMIIA KD and NMIIB KD cells respectively, from four independent experiments. (f–h) Mann-Whitney statistical tests were applied for p value. (i) Representative spinning disk GFP-E-cadherin image sequences over a time course of 4 hr showing the dynamics of junction formation at high magnification in Ctrl, NMIIA KD and NMIIB KD MDCK cells. The red arrows point at junctional extensions typically observed in NMIIB KD doublets. Scale bar: 10 μm.

DOI: https://doi.org/10.7554/eLife.46599.008

The following source data and figure supplement are available for figure 2:

**Source data 1.** NMIIA and NMIIB are both required for proper junction biogenesis.
DOI: https://doi.org/10.7554/eLife.46599.010
**Figure supplement 1.** Isoform-specific NMII Knock-down in MDCK cells.
DOI: https://doi.org/10.7554/eLife.46599.009

colleagues (*Menko et al., 2014*) in lens epithelial cells. Importantly, the localization of each isoform was not affected by the silencing of the other isoform (*Figure 3—figure supplement 1c*).

NMIIA and NMIIB were previously reported to localize to apical epithelial junctions in polarized MCF-7 cells (*Smutny et al., 2010*; *Gomez et al., 2015*) with however some divergencies. Thus, we followed the localization of both isoforms during apico-basal polarization of MDCK cells (*Figure 3e– g* and *Figure 3—figure supplement 2a*). After one day of culture, NMIIB and NMIIA were differentially localized in sub-confluent cell clusters. NMIIA was associated to stress fibres and excluded from junctional membranes while NMIIB colocalized with E-cadherin at cell-cell contacts. After 3 days of culture, confluent MDCK cells started to develop an apico-basal polarization and the two isoforms associated to apically positioned *zonulae adherens*. However, even at these stages, only NMIIB colocalized with E-cadherin, while NMIIA was accumulated perijunctionnally as previously reported in MCF-7 cells (*Gomez et al., 2015*). At the basal side, they were both associated to stress fibres. To confirm the differential localization of the two isoforms we analysed the distribution in transiently transfected MDCK cells of GFP-NMIIA and of mCherry-NMIIB (*Figure 3—figure supplement 1d,e*). mCherry-NMIIB accumulated at junctional membranes while GFP-NMIIA accumulated in perijunctional areas as reported in *Ozawa (2018)*. These differential distributions at the early stages of AJ formation were not specific to MDCK cells, and were observed as well in small clusters of Caco2 cells (*Figure 3—figure supplement 2b*). Considering recent findings showing a possible interaction between NMIIB and α-catenin (*Vassilev et al., 2017*), we hypothesized that NMIIB could be recruited to the junction through α-catenin/E-cadherin complexes. Accordingly, in α-catenin KD MDCK cells (*Benjamin et al., 2010*), NMIIB was relocalized to NMIIA-enriched stress fibres and circumnuclear actin cables (*Figure 3—figure supplement 2c,d*), indicating that α-catenin is required for NMIIB junctional recruitment.

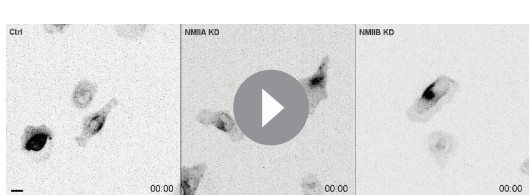

**Video 3.** Dynamic of junction formation in Ctrl, NMIIA KD and NMIIB KD cells. Epi-fluorescence movies of Ctrl, NMIIA KD and NMIIB KD MDCK cells expressing GFP-E-cadherin. Scale bar: 10 μm.
DOI: https://doi.org/10.7554/eLife.46599.011

To better characterize the organization of the actomyosin cytoskeleton at nascent AJs, co-stainings of NMIIA, NMIIB, F-actin and β−catenin performed on control MDCK cells were imaged using structured illumination microscopy (SIM). NMIIA was associated to thick F-actin bundles running parallel to, and located a few microns away from the junctional membranes (*Figure 4a,c,d*), as reported for NMIIA localization in linear junctions of endothelial cells (*Hoelzle and Svitkina, 2012*; *Efimova and Svitkina, 2018*). We confirmed at this resolution that NMIIA did not colocalize with β−catenin-labeled cadherin-catenin complexes. Interestingly, NMIIA appeared distributed on actomyosin bundles in sarcomere like structures as described before in other cellular contexts (*Choi et al., 2016*; *Ebrahim et al., 2013*). NMIIB junctional staining colocalizing with β−catenin was associated with a 200 nm to 1 μm thick fuzzy F-actin network (*Figure 4a–d*), that also contained both Arp2/3 (*Figure 3—figure supplement 2e*) and cortactin (*Figure 5b–d*), two known molecular markers of branched actin meshwork. Looking at short junctions that probably corresponded to nascent cell-cell contacts, we could also observe the strong enrichment of NMIIB and the exclusion of NMIIA at the contact zone (*Figure 4e,f*).

Altogether, these observations reveal that early during AJ biogenesis, NMIIB is associated to a juxtamembrane actin meshwork, structurally distinct from the perijunctional contractile actin bundles running parallel to the junction where NMIIA is preferentially associated.

## NMIIA regulates the organization of perijunctional actin bundles while NMIIB regulates the organization of a juxtamembrane actin layer

Based on these observations and previous studies (*Smutny et al., 2010*; *Efimova and Svitkina, 2018*), we subsequently explored the possibility that NMIIB and NMIIA could differentially regulate actin organization at the junction, thereby maintaining its structural integrity. Using SIM microscopy, we analyzed the organization of junctional actin cytoskeleton in NMIIA KD and NMIIB KD cells. NMIIA KD cells exhibited shorter actin bundles running parallel to the junction, while their juxtamembrane F-actin meshwork was comparable to the one of Ctrl cells, both in terms of morphology and cortactin staining (*Figure 5a,e,f* and *Figure 5—figure supplement 1a,b*). In contrast, NMIIB KD cells presented a strongly enlarged area of junctional F-actin meshwork colocalizing with β−catenin that corresponded to overlapping membrane extensions stained with cortactin (*Figure 5a,e,f* and *Figure 5—figure supplement 1a,b*). In addition, while they retained some of the perijunctional actin bundles, we could observe numerous oblique actin bundles directed toward the junction (*Figure 5a* and *Figure 5—figure supplement 1a,b*). These results show that NMIIA supports the organization of perijunctional actin bundles while NMIIB contributes to restrain the extent of the juxtamembrane F-actin meshwork that couples perijunctional bundles to the plasma membrane, thus restraining lamellipodial activity at the junction.

An Arp2/3-nucleated actin network at the *zonula adherens* has been shown to regulate junctional tension in epithelial monolayers (*Verma et al., 2012*). On the other hand, junctional tension has been shown to associate with the presence of α-catenin molecules under open conformations (*Ishiyama et al., 2018*; *Yonemura et al., 2010*). Moreover, a direct link between α−catenin and NMIIB has been reported (*Vassilev et al., 2017*), suggesting that NMIIB recruitment, α-catenin molecular unfolding and regulation of branched actin polymerization could be tightly linked. Thus, we performed immunostainings with the α18 monoclonal antibody recognizing the open conformation of the protein (*Yonemura et al., 2010*). Strikingly, the ratio of α18 on total α−catenin junctional staining was decreased by four times in NMIIB KD cells compared to Ctrl cells, while it was not affected in NMIIA KD cells. This suggests that junctional α-catenin molecules were significantly turned to the closed conformational state in NMIIB KD cells (*Figure 5g–i*). In contrast, the total α-catenin junctional levels were significantly reduced in NMIIA KD cells, as shown by others (*Shewan et al., 2005*; *Smutny et al., 2010*). Taken together, these results strengthen complementary contributions for NMIIB and NMIIA where NMIIB is the main isoform required for the organization of juxtamembrane actin cushion and NMIIA for organization of perijunctional contractile actin fibres.

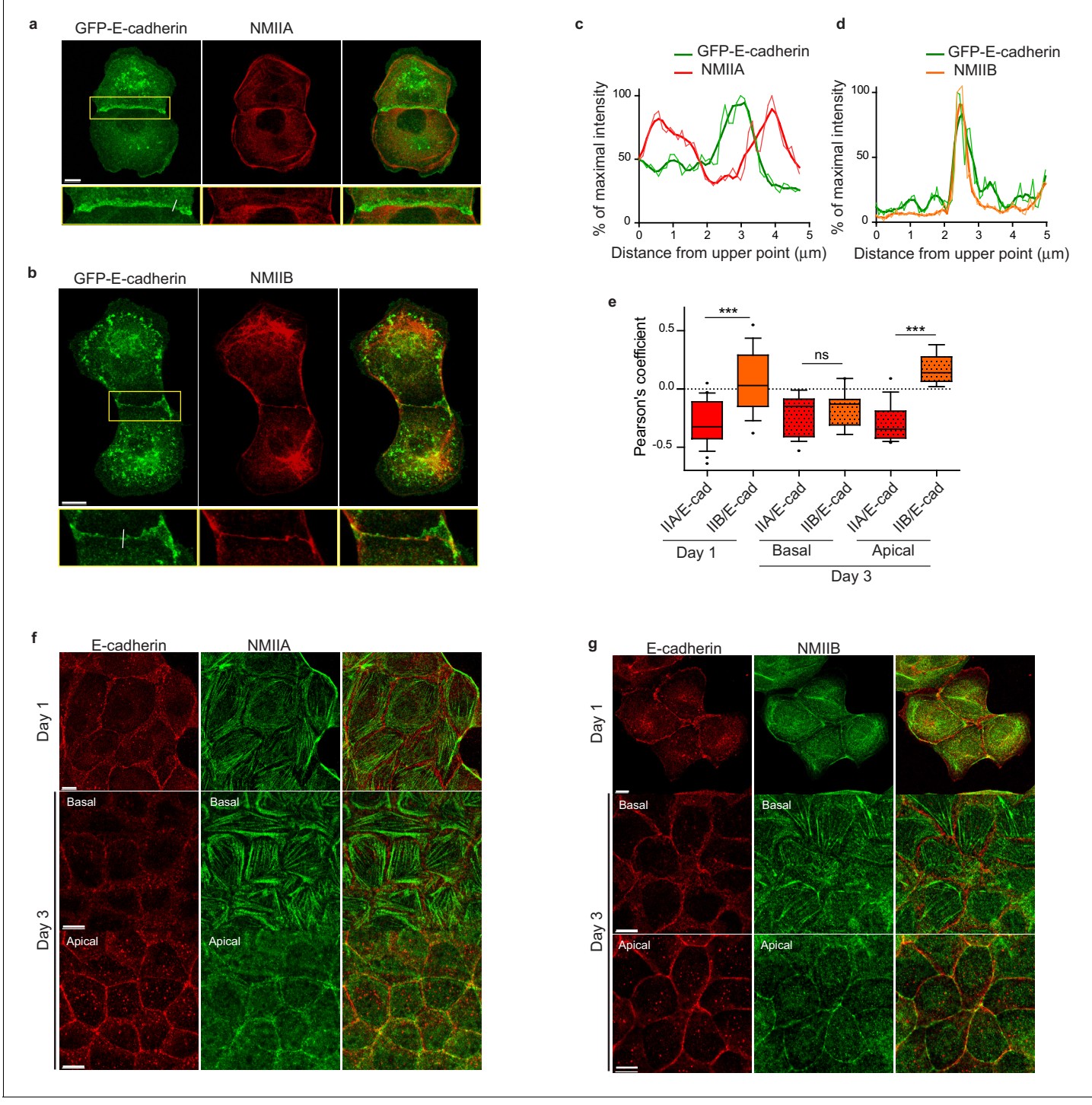

**Figure 3.** NMIIB, but not NMIIA, localizes to early AJs. (**a, b**) Representative confocal images and zoom boxes of GFP-E-cadherin-expressing MDCK cell doublets fixed 20 hr after BCN-RGD addition and immuno-stained for NMIIA (**a**) or NMIIB (**b**) Scale bar: 10 μm. (**c, d**) Relative intensity profiles (raw and smoothed data) of GFP-E-cadherin and NMIIA (**c**) or NMIIB (**d**) signals along the lines represented in (**a**) and (**b**) respectively. (**e**) Box and whiskers graphs showing the Pearson's coefficient values that reflects the co-localization of NMIIA, NMIIB with E-cadherin quantitatively. n = 9 to 24 junctions. Mann-Whitney statistical tests were applied for p value. (**f, g**) Representative confocal images of WT MDCK cells plated on fibronectin-coated glass for 1 or 3 days and stained for F-actin, NMIIA (**f**) and NMIIB (**g**). Scale bar: 10 μm.

DOI: https://doi.org/10.7554/eLife.46599.012

The following source data and figure supplements are available for figure 3:

**Source data 1.** NMIIB, but not NMIIA, localizes to early AJs.

*Figure 3 continued on next page*

*Figure 3 continued*

DOI: https://doi.org/10.7554/eLife.46599.017

**Figure supplement 1.** NMIIA and NMIIB exhibit differential localizations in early AJs.

DOI: https://doi.org/10.7554/eLife.46599.013

**Figure supplement 1—source data 1.** NMIIA and NMIIB exhibit differential localizations in early AJs.

DOI: https://doi.org/10.7554/eLife.46599.014

**Figure supplement 2.** NMIIB, but not NMIIA, localizes to early epithelial AJs.

DOI: https://doi.org/10.7554/eLife.46599.015

**Figure supplement 2—source data 1.** NMIIB, but not NMIIA, localizes to early epithelial AJs.

DOI: https://doi.org/10.7554/eLife.46599.016

## NMIIA is required for the generation of forces at E-cadherin adhesions while NMIIB favors their transmission through F-actin anchoring

The formation of cell-cell junctions in cell doublets is concomitant with the formation of cell-matrix adhesions and the tugging force applied on cell-cell contacts must be compensated by traction of the cells on cell-matrix adhesion complexes (*Liu et al., 2010*; *Maruthamuthu et al., 2011*; *Ng et al., 2014*). To further understand the contributions of NMII isoforms in junction biogenesis, we thus experimentally decoupled these two adhesion systems. We first investigated the role of NMII isoforms in cell-matrix adhesion by seeding single Ctrl, NMIIA KD and NMIIB KD cells on fibronectin-coated glass. NMIIA KD cells spread 1.7 times more than Ctrl and NMIIB KD cells on fibronectin and their actin cytoskeleton was highly perturbed exhibiting a strong decrease in ventral stress fibres and cortical actin bundles together with an enlargement of their lamellipodia (*Figure 6—figure supplement 1a,b*). NMIIA KD cells also formed significantly less focal adhesions (*Figure 6—figure supplement 1a,c*). In contrast, NMIIB KD cells showed no defect in actin organization, cell spreading or focal adhesion formation (*Figure 6—figure supplement 1a–c*). Next, we measured by TFM the magnitude of traction forces applied by single cells on deformable fibronectin-coated 30 kPa PDMS gels. NMIIA KD cells exerted lower traction forces than Ctrl cells as reported by others (*Jorrisch et al., 2013*; *Shutova et al., 2017*). NMIIB KD cells, on the contrary, did not show any defect in traction force generation on this substratum (*Figure 6—figure supplement 1d,e*). These results, in agreement with previous studies (*Jorrisch et al., 2013*; *Sandquist et al., 2006*), show that NMIIA is the isoform regulating cell spreading, cell adhesion, traction force generation and organization of contractile actin structures on fibronectin. In contrary MNIIB is not contributing at all to the cell-matrix adhesion, focal adhesion formation, actomyosin reorganization and traction forces on fibronectin.

To explore the contribution of NMII isoforms to E-cadherin-mediated cell-cell adhesion per se, we seeded single cells on E-cadherin-coated substrates (*Figure 6—figure supplement 2a,b*). After 6 hr, Ctrl and NMIIA KD cells had spread similarly with mean areas of 1178 ± 40 µm² and 1031 ± 37 µm² respectively, while NMIIB KD cell spreading was significantly reduced (mean area = 515 ± 21 µm²) (*Figure 6—figure supplement 2a,c*). Ctrl cells organized thick circumnuclear actin arcs, as well as radial actin fibres connected to peripheral β-catenin clusters (*Figure 6—figure supplement 2a*), as previously described (*Gavard et al., 2004*; *Collins et al., 2017*). NMIIA KD cells, while spreading as Ctrl cells on E-cadherin lacked the circumnuclear actin arcs and formed fewer large and small cadherin clusters (*Figure 6—figure supplement 2a,d*). In particular they could not organize large clusters aligned along actin cables, as reported in MCF7 cells (*Smutny et al., 2010*). NMIIB KD cells kept the organization of circumnuclear actin arcs, but were depleted of radial actin bundles, did not form significant β-catenin clusters and failed to spread on E-cadherin (*Figure 6—figure supplement 2a,c,d*). Contrasting with data obtained in MCF7 cells (*Smutny et al., 2010*), these data indicated that NMIIB plays a major role in the clustering and stabilization of E-cadherin/catenin complexes that in turn promote cell spreading. Our findings also suggest that NMIIA is required for the formation of contractile actin fibres that apply traction forces on the cadherin adhesions. We thus measured the capacity of these cells to transmit forces through E-cadherin complexes by TFM, seeding them on E-cadherin-coated 15 kPa PDMS elastic gels. Compared to Ctrl cells, NMIIA KD cells exhibited very low forces on E-cadherin substrate (*Figure 6—figure supplement 2e,f*), confirming that NMIIA generates the forces transmitted to E-cadherin adhesions. NMIIB KD cells, that failed to cluster cadherin/catenin complexes, also generated lower traction forces than Ctrl cells, albeit to a lesser

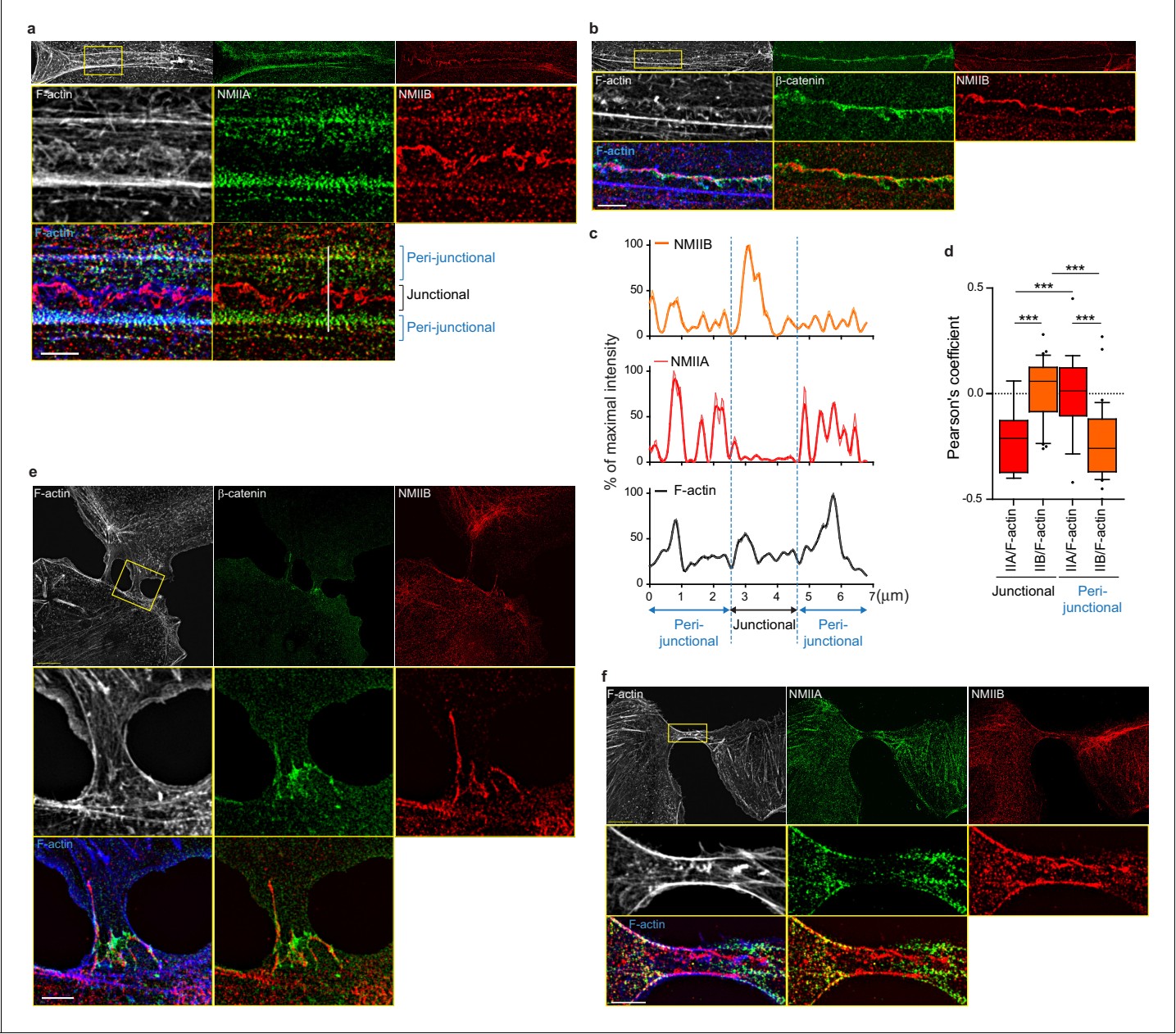

**Figure 4.** NMIIB localizes to a junctional actin network distinct from NMIIA-associated actin. (**a–b**) SIM (Structured Illumination Microscopy) images of WT MDCK cells fixed 20 hr after addition of BCN-RGD and stained as indicated. Scale bar: 3 µm. (**c**) Relative intensity profiles (raw and smoothed data) of NMIIB, NMIIA and F-actin signals along the line represented in (**a**). (**d**) Box and whiskers graphs showing the Pearson's coefficient values that reflects the co-localization of F-actin and NMIIA or NMIIB in junctional and peri-junctional areas. n = 18 to 33 junctions. For p values, pairwise t tests were applied to compare junctional vs perijunctional data for the same isoform and Mann-Whitney statistical tests to compare the two isoforms. (**e, f**). SIM images of nascent contacts formed between WT MDCK cells. Scale bar: 3 µm.

DOI: https://doi.org/10.7554/eLife.46599.018

The following source data is available for figure 4:

**Source data 1.** NMIIB localizes to a junctional actin network distinct from NMIIA-associated actin.

DOI: https://doi.org/10.7554/eLife.46599.019

extent than NMIIA KD cells (***Figure 6—figure supplement 2e,f***). Even though both NMII isoforms contribute to cell-generated forces on E-cadherin substratum, they have complementary contributions. NMIIA is required for the formation of stress fibres while NMIIB would rather regulate the

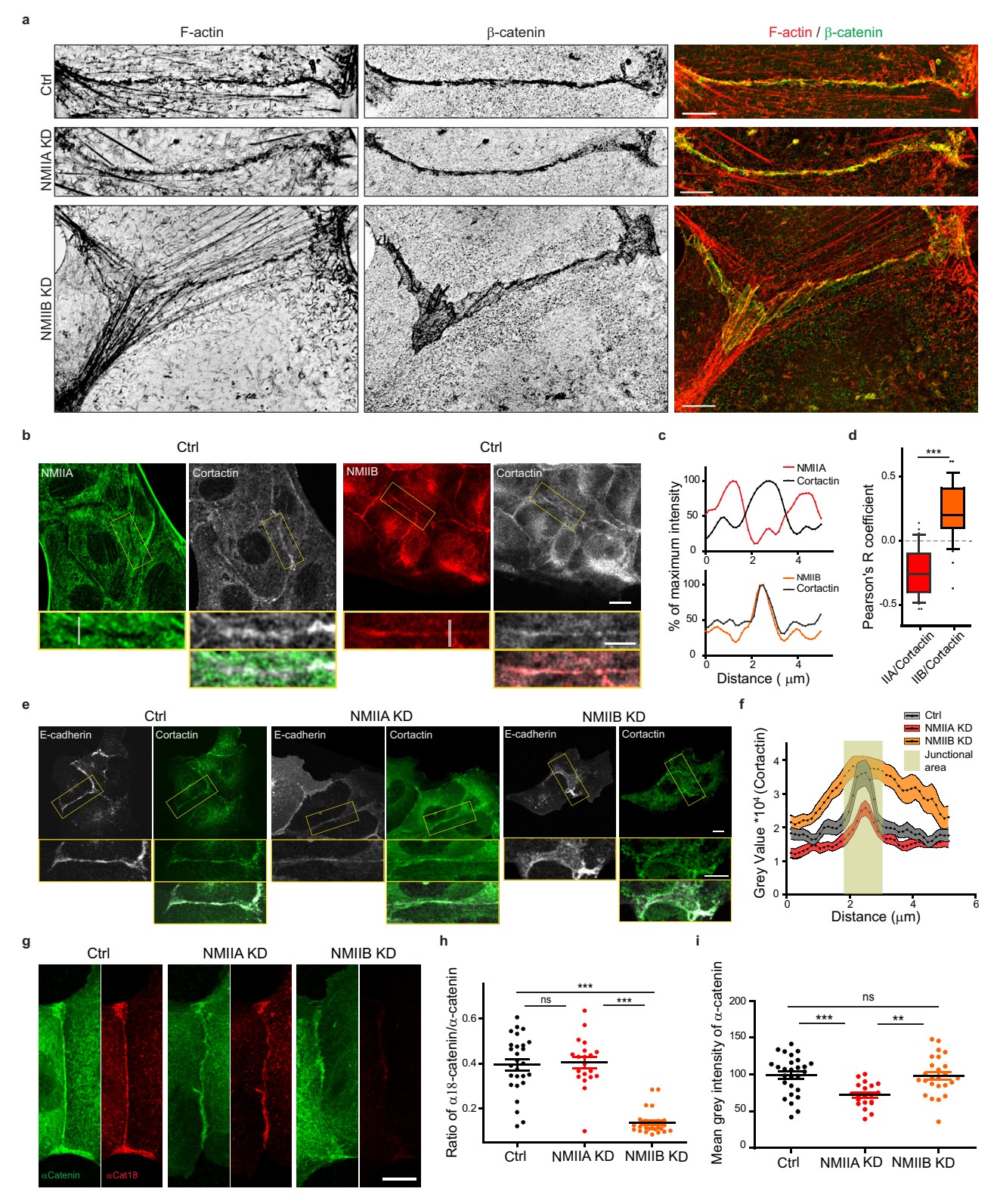

**Figure 5.** NMIIB supports juxtamembrane actin organization and regulates α-catenin unfolding. (a) SIM (Structured Illumination Microscopy) images of junctional areas from Ctrl, NMIIA KD and NMIIB KD cells fixed 20 hr after addition of BCN-RGD and stained for F-actin and β-catenin. Scale bar: 5 μm. (b) Representative confocal images with zoom boxes of Ctrl MDCK cells. (c) Relative intensity profiles of cortactin and NMIIA or NMIIB signals along the lines represented in (b, d). Box and whiskers graphs showing the Pearson's coefficient values for co-localization of cortactin with NMIIA or NMIIB at cell-

*Figure 5 continued on next page*

*Figure 5 continued*

cell junctions n = 31 and 36 junctions respectively, Mann-Whitney statistical tests were applied for p value. (**e**) Ctrl, NMIIA KD and NMIIB KD cells stained for NMIIA, NMIIB and cortactin as indicated. Scale bars: 10 µm in original and 5 µm in zoomed images. (**f**) Relative intensity distribution profiles of cortactin signal along lines drawn perpendicular to junction in Ctrl, NMIIA KD and NMIIB KD cells, n = 15 cell-cell junctions respectively. (**g**) Representative confocal images of junctional area from Ctrl, NMIIA KD and NMIIB KD cells stained for α-catenin and α-cat18. Scale bar: 10 µm. (**h, i**) Scatter plots with mean + /- SEM showing the ratio of junctional α-cat18/α-catenin signals (**h**) and the mean intensity levels of α-catenin signal at the junction (**i**) n = 27, 20, 25 cell doublets for Ctrl, NMIIAKD and NMIIBKD, respectively from two independent experiments. Kruskal-Wallis statistical tests were applied for p value.

DOI: https://doi.org/10.7554/eLife.46599.020

The following source data and figure supplement are available for figure 5:

**Source data 1.** NMIIB supports juxtamembrane actin organization and regulates α-catenin unfolding.

DOI: https://doi.org/10.7554/eLife.46599.022

**Figure supplement 1.** NMIIB supports junctional actin organization.

DOI: https://doi.org/10.7554/eLife.46599.021

transmission of force and the coupling of actin stress fibres to the cadherin-catenin complexes. Slight divergences on localizations and effects of silencing reported here compared to data obtained in MCF7 cells (*Smutny et al., 2010*; *Gomez et al., 2015*) may relate to the reported changes in homo/heteropolymerization of NMIIA and NMIIB (*Shutova et al., 2014*; *Beach et al., 2014*) which could depend on relative levels of expression and/or maturation of contractile actin fibres.

## NMIIA and NMIIB are required for proper organization of inter-cellular junctional stress

To directly determine how NMIIA and NMIIB contribute to traction force generation and transmission during AJ biogenesis, we mapped traction forces before and after cell-cell contact formation in cell doublets. Hotspots of traction forces were generated at the periphery of the doublet where lamellipodia arise (*Figure 6a*). As expected from the TFM data obtained with single cells seeded on fibronectin, NMIIA KD doublets, compared to Ctrl and NMIIB KD ones, exhibited lower traction forces both before and after cell-cell contact formation (*Figure 6a–c*). NMIIB KD doublets developed traction forces similar in magnitude to those developed by Ctrl ones, with however different patterns. Hotspots of forces frequently appeared in the junctional area in NMIIB KD doublets that were generally absent in Ctrl and NMIIA doublets (*Figure 6a*). We quantified these differences by analysing the spatial repartition of forces in the peripheral and central subdomains of the junction, and their orientation relative to main junction axis (parallel, $F_{//}$, and perpendicular, $F_{\perp}$, components). NMIIB KD doublets generated higher $F_{\perp}$ in the central part of the junction and lower values of $F_{//}$ with respect to Ctrl doublets in both the peripheral and the central part (albeit not significantly) of the junction (*Figure 6—figure supplement 3a,b*). As a consequence, the ratios of parallel/perpendicular forces in the central and peripheral part of the junction were lower in NMIIB KD doublets compared to Ctrl doublets (*Figure 6—figure supplement 3c*). The mechanical perturbation induced by knocking down NMIIB leads to a redistribution of transmitted forces in the junctional area at both cell-cell and cell-substrate interfaces. These results show that NMIIB plays an important role in the repartition of traction forces under the junction and that NMIIA is essential for the generation of traction forces in general. We next quantified the capacity of NMIIA KD and NMIIB KD cells to transmit forces across the junction. Following Newton's laws, the net traction force exerted by an isolated doublet is zero, up to the measurement noise. Conversely, the net traction forces exerted by each of the two cells are equal in magnitude and opposite in direction, compensating exactly (*Liu et al., 2010*; *Maruthamuthu et al., 2011*; *Ng et al., 2014*). We thus calculated the resultant vectorial sum of forces per cell (*Figure 6b*). In all conditions, the resultant force per cell before contact was within the level of noise as expected for isolated cells and increased within 30 min after contact to reach a plateau, attesting the capacity of all three cell lines to transmit intercellular tugging forces across the junction (*Figure 6b,c*). However, in NMIIA KD cells, the resultant forces per cell at the plateau was significantly lower than in Ctrl and NMIIB KD cell doublets (*Figure 6c*), which is consistent with the inability of these cells to apply strong traction forces on fibronectin substratum.

Using traction force measurement data, we then computed the intracellular stress in the cell doublets (*Nier et al., 2016*) (*Figure 6a*). The in-plane stress is represented by a tensor with three

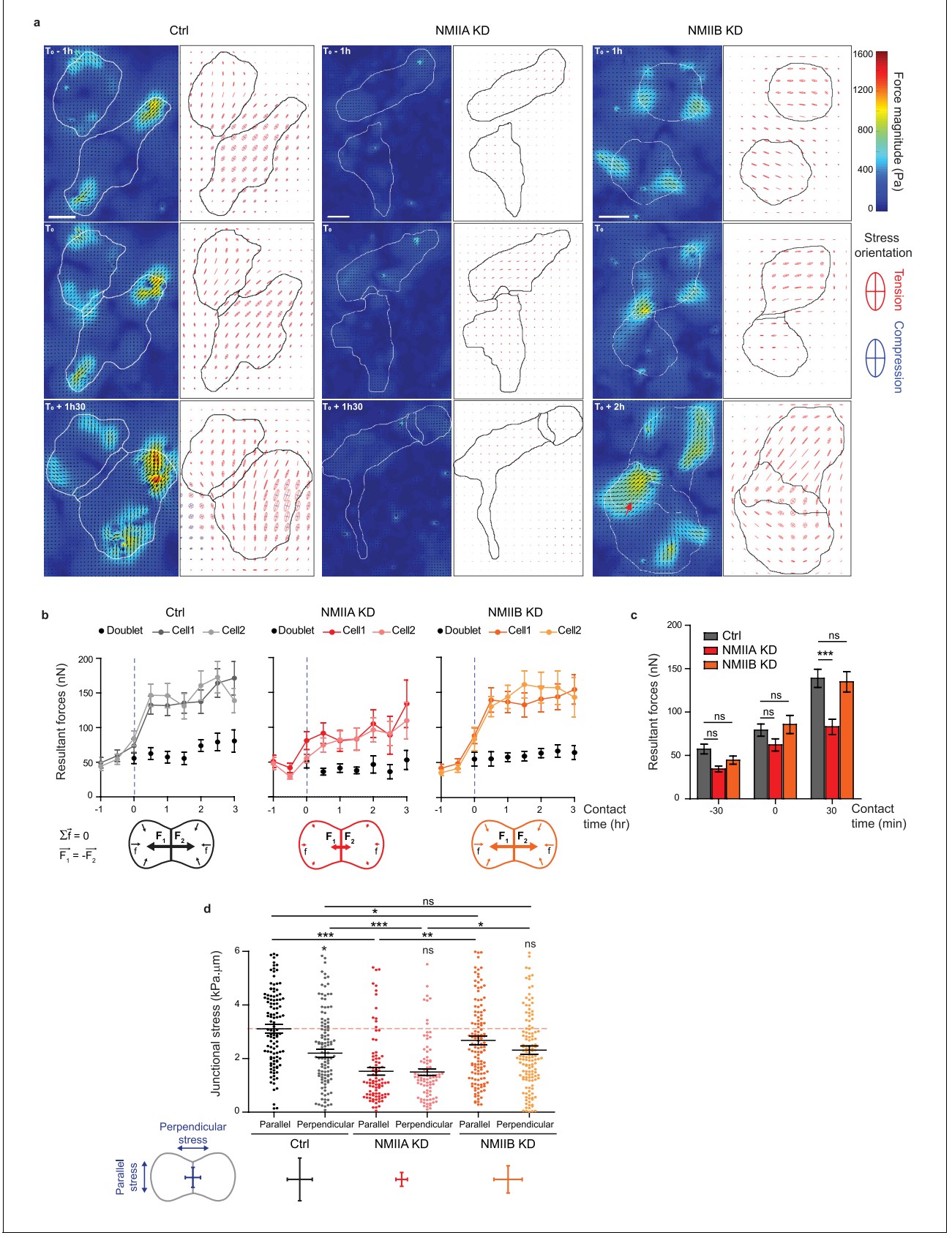

**Figure 6.** NMIIA and NMIIB are both required for establishment of proper inter-cellular stress. (a) Heat map with vectorial field of traction forces (left panels) and ellipse representation of intra-cellular stress (right panel, the two axes represent the direction and magnitude of the principal components of the stress tensor, positive values in red, negative values in blue) of inter-cellular stress (right panels) in Ctrl, NMIIA KD and NMIIB KD cell pairs before, during and after contact on fibronectin-coated PDMS deformable substrate (30 KPa). Cell contours are drawn in black. The red arrows indicate a hotspot of traction forces observed frequently in NMIIB KD cell doublets. Scale bar: 10 μm. (b) Linear graphs representing the resultant forces of cell doublets and individual cells before, during and after contact in Ctrl, NMIIA KD and NMIIB KD. Data are represented as mean + /- SEM. (c) The same data as in (b) were represented as bar graph with mean + /- SEM for statistical comparisons between Ctrl, NMIIA KD and NMIIB KD cells 30 min before, during and 30 min after contact. Bonferroni statistical tests were applied for p value. (d) Scatter plots with mean + /- SEM representing inter-cellular stress in the junctional area in Ctrl, NMIIA KD and NMIIB KD cells within the first 3 hr of contact. For each junction, six values corresponding to 30 min time points are plotted. The stress orientation was divided in the parallel and perpendicular components relative to the main axis of the junction. Pairwise statistical t tests (for intra-group comparisons) and Mann-Whitney statistical t tests were applied for p value. (b–d) n = 25, 26 and 28 cell doublets for Ctrl, NMIIA KD and NMIIB KD, respectively, from three independent experiments.

DOI: https://doi.org/10.7554/eLife.46599.023

The following source data and figure supplements are available for figure 6:

**Source data 1.** NMIIA and NMIIB are both required for establishment of proper inter-cellular stress.
DOI: https://doi.org/10.7554/eLife.46599.030
**Figure supplement 1.** NMIIA regulates cell adhesion and traction forces on fibronectin.
DOI: https://doi.org/10.7554/eLife.46599.024
**Figure supplement 1—source data 1.** NMIIA regulates cell adhesion and traction forces on fibronectin.
DOI: https://doi.org/10.7554/eLife.46599.025
**Figure supplement 2.** NMIIB favors E-cadherin clustering on E-cadherin-coated substrate.
DOI: https://doi.org/10.7554/eLife.46599.026
**Figure supplement 2—source data 1.** NMIIB favors E-cadherin clustering on E-cadherin-coated substrate.
DOI: https://doi.org/10.7554/eLife.46599.027
**Figure supplement 3.** NMIIA and NMIIB are both required for establishment of proper inter-cellular stress.
DOI: https://doi.org/10.7554/eLife.46599.028
**Figure supplement 3—source data 1.** NMIIA and NMIIB are both required for establishment of proper inter-cellular stress.
DOI: https://doi.org/10.7554/eLife.46599.029

independent components: two components of normal stress denoting either tension (positive values) or compression (negative values) along the corresponding directions, and one component of shear stress, except in the basis of the tensor's principle directions, where there is no shear stress. The ellipse representation in *Figure 6a* shows that the stress is highly anisotropic, and the cells are mostly under tension except for regions of very small compression associated to high tension in the other direction. The NMIIA KD cells show lower tension, consistent with the lower amount of traction forces they exert. We focused on the normal stress within the region of cell-cell junction, as AJs provide a mechanical link that drives transmission of forces between cells and thus organize inter-cellular stress (*Nier et al., 2016*; *Saw et al., 2017*). We thus computed the perpendicular ($\sigma_\perp$) and parallel ($\sigma_{//}$) components of normal stress relative to the junction axis, which characterize the tension across and along the junction respectively. Within 30 min after contact formation, the junction was submitted to a rise of $\sigma_\perp$ in all three cell lines, consistently with the emergence of a cell-cell tugging force (*Figure 6—figure supplement 3d,e*). However, in Ctrl cells, the normal stress parallel to the junction $\sigma_{//}$, remained higher than $\sigma_\perp$ (*Figure 6d*). Strikingly, this was not the case in NMIIB KD and NMIIA KD cells that exhibited equal amounts of normal stress parallel and perpendicular to the junction, denoting a more isotropic distribution of junctional tension (*Figure 6d*).

Altogether, these results show that NMIIA and NMIIB are both required for mechanical integrity of the junction. NMIIA is necessary for generation of a high junctional inter-cellular stress through production of tugging forces compensated by traction applied at cell-matrix adhesions. NMIIB, on the other hand, is necessary for the establishment of an anisotropic stress at the junction, sustaining high tension along the cell-cell interface.

## Discussion

Here, we explore for the first time the involvement of NMII isoforms during early steps of epithelial junction formation. We show that NMIIA and NMIIB associate with distinct pools of actin and

cooperate to initiate the formation of epithelial AJ before the acquisition of the apico-basal polarization (See *Figure 7*).

Our careful examination by SIM during junction biogenesis revealed precise patterns of NMII, actin and E-cadherin localization whereas other studies mostly focused on actin and NMII in mature junctions (*Smutny et al., 2010*; *Gomez et al., 2015*). While NMIIA associated to actin bundles parallel to- and distant from the junction, NMIIB was sitting at junctional membranes in association with a juxtamembrane actin network, distinct from NMIIA-associated actin. The existence of two distinct actin networks at *adherens* junctions had already been observed in early junctions between hepatocytes (*Krendel and Bonder, 1999*) and in endothelial cells where VE-cadherin was shown to colocalize with Arp2/3 complex-positive actin networks in-between distal actin-NMII bundles (*Efimova and Svitkina, 2018*). The localization of NMIIA is reminiscent of what has been observed previously in linear AJ of endothelial cells (*Efimova and Svitkina, 2018*). Strikingly, we show here an unexpected association of NMIIB with a juxtamembrane actin cushion that links the junctional membrane to NMIIA-associated perijunctional contractile actin bundles. Our data support a role of NMIIB in linking adhesion complexes and perijunctional actin bundle and on restraining lamellipodial activity at the junction. We believe that these are properties common to the early stage of AJ formation in many cell types that then mature to elaborate *zonulae adherens* in epithelial cells where both actin organizations persist but become tightly packed to the junctional membrane. Interestingly, in the absence of α-catenin, the localization of NMIIB was not restricted any more to junctional membranes. Instead, NMIIB co-assembled with NMIIA on the same actin fibres, likely in heretotypic minifilaments, as observed in previous studies (*Beach and Hammer, 2015*; *Shutova et al., 2017*), indicating that α-catenin is responsible for the junctional recruitment of NMIIB, as reinforced by a recent publication reporting NMIIB and α-catenin interaction in glioblastoma cells (*Vassilev et al., 2017*).

These distinct localization patterns at early junctions are correlated to differential contributions of NMIIA and NMIIB in junction biogenesis. Upon contact formation, NMIIA KD cells were unable to elongate the junction and to sustain long-lived cell-cell contacts. They also lacked the capacity to produce traction forces on E-cadherin-coated substrates. Our observations thus identify NMIIA as the major isoform responsible for the NMII-dependent mechanical tugging force required for junction growth (*Liu et al., 2010*). This was confirmed by traction force and stress analysis data revealing a decrease of the forces as well as a reduction of both parallel and perpendicular stresses at the junction for NMIIA KD cells. In contrast, NMIIB KD cells transmitted elevated tugging forces and maintained cell-cell contacts, but their junctions appeared enlarged and twisted with a lower parallel stress. These results are remarkable given that NMIIB was found to be expressed 100 times less than NMIIA in MDCK cells (*Ma et al., 2010*). NMIIB was required for efficient E-cadherin clustering on E-cadherin substrates and for the connection of the contractile actin network to these clusters. NMIIB was required for the proper organization and spatial restriction of the juxtamembrane actin network and was also the main isoform responsible for the maintenance of α-catenin in an opened conformation. However, we observed a reduced junctional recruitment of α-catenin in NMIIA KD cells, suggesting also a contribution of NMIIA in α-catenin activation in agreement with a previous report (*Ozawa, 2018*).

Given that E-cadherin complexes have been shown to biochemically interact with both Arp2/3 (*Kovacs et al., 2002*; *Verma et al., 2012*) and NMIIB (*Vassilev et al., 2017*), one hypothesis could be that NMIIB and Arp2/3 are both recruited to E-cadherin/catenin complexes upon cell-cell contact initiation. NMIIB could thus serve as a cross-linker of the junctional actin network. Hence, the absence of NMIIB may keep α-catenin in a closed conformation and induce a local softening of AJs which in turn leads to increased junctional lamellipodial extension. It is also in agreement with a previous study showing that Arp2/3-nucleated actin network at the *zonula adherens* regulates junctional tension and integrity (*Verma et al., 2012*). NMIIB, by associating both with cadherin-catenin complexes and the branched actin, could somehow rigidify and regulate the thickness of this F-actin cushion sitting between the membrane and the contractile actin fibres associated to NMIIA. This could be achieved through the specific biochemical properties of NMIIB towards actin that provide it with the capacity to transmit tension within actin filaments at low energetic cost (*Kovács et al., 2007*; *Ma et al., 2007*; *Ma et al., 2012*). Along this line, it is striking to note that we never observe in early AJ any sign of organization of NMIIB in minifilaments in the junctional area as observed for NMIIA in perijunctional actin bundles.

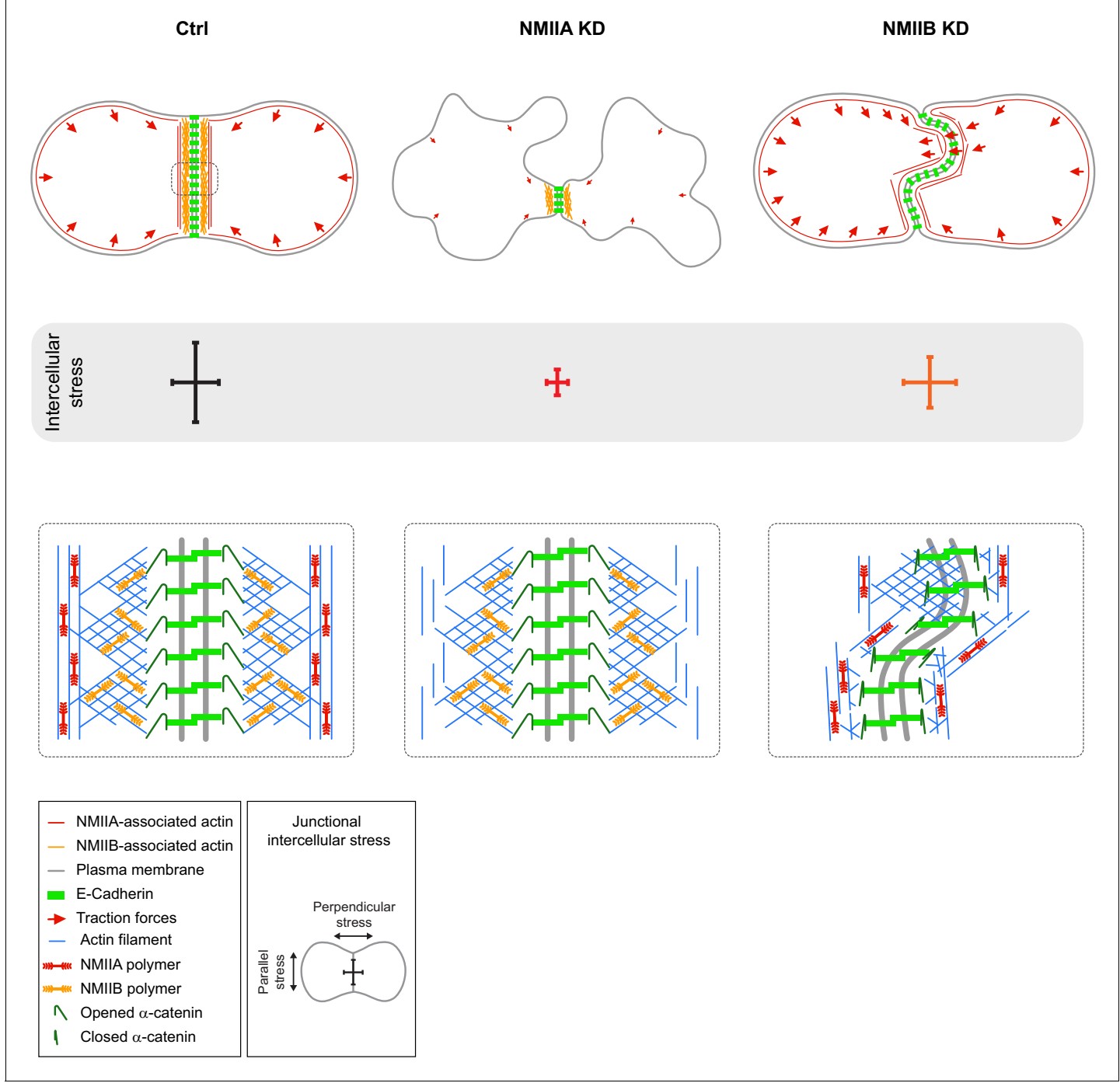

**Figure 7.** Proposed model for the role of NMIIA and NMIIB during junction biogenesis. Upper panels: organization of early cell-cell contacts of Ctrl, NMIIA KD and NMIIB KD cells. Lower panels: proposed molecular organization of early junctions. Middle panels: distribution of intercellular stress. Ctrl cells establish stable and straight junctions maintained under an anisotropic intercellular stress preeminent parallel to the junction. NMIIB associates to- and organizes the junctional branched actin meshwork. NMIIA, which provides mechanical tugging force, sits on distant perijunctional actin bundles parallel to the junction. NMIIA KD cells fail to maintain stable cell-cell contacts exhibit shorter junctions, weak traction forces and weak intercellular stress. Perijunctional actin bundles are smaller and disorganized. NMIIB KD cells establish persistent but wavy junctions from which lamellipodial extensions and traction force hotspots arise. The junctional branched actin meshwork is disorganized which probably prevents α-catenin opening and induces the formation of lamellipodial extensions. The anchoring of perijunctional actin bundles to the junction is perturbed, despite the presence of NMIIA. There is, in these cells, no preferential orientation of intercellular stress.

DOI: https://doi.org/10.7554/eLife.46599.031

Inter-cellular stress is generated at cell-cell adhesions, although this remained poorly characterized (*Maruthamuthu et al., 2011*; *Ng et al., 2014*). Here, we evaluated the amount and the orientation of intercellular stress generated during junction biogenesis. Within one hour of cell-cell contact, an anisotropic intercellular stress appeared at the junction, with a preferential orientation parallel to the junction, favoring the elongation and the stability of the nascent junction. Both isoforms were required for proper establishment and orientation of this intercellular stress. NMIIA silencing had a global impact on the amount of intercellular stress generated, which was not surprising given its role on traction force production both at cell-matrix and cell-cell adhesions. On the other hand, NMIIB favored the production of a higher parallel intercellular stress, probably by driving the crosslinking and stiffening of the junctional actin network that couples the perijunctional contractile actin to the plasma membrane.

In conclusion, we demonstrate here that both NMIIA and NMIIB contribute to the early steps of AJ biogenesis and are necessary for mechanical integrity of the junction, albeit implicated in very different aspects of adhesion complexes and actin pools organization. These findings open new avenues in the understanding of how distinct pools of actomyosin, associated to different myosin isoforms, build up and integrate mechanical forces to regulate adherens junction remodeling and intercellular stress in vertebrate cells in order to achieve large scale tissue remodeling during embryogenesis and tissue repair.

# Materials and methods

**Key resources table**

| Reagent type (species) or resource | Designation | Source or reference | Identifiers | Additional information |
|---|---|---|---|---|
| Cell line (*Canis familiaris, dog*) | MDCK | ATCC | ATCC CCL-34 | |
| Cell line (*H. sapiens*) | Caco-2 | ATCC | ATCC HTB-37 | Kindly provided by S.Robine (Institut Cuire/CNRS, Paris) |
| Antibody | anti-NMIIA rabbit polyclonal | Biolegend | 909801 | 1/100 for IF and 1/1000 for WB |
| Antibody | anti-NMIIA mouse monoclonal | Abcam | ab55456 | 1/100 for IF and 1/1000 for WB |
| Antibody | rabbit anti-NMIIB polyclonal | Biolegend | 909901 | 1/100 for IF and 1/1000 for WB |
| Antibody | anti-β-catenin rabbit polyclonal | Sigma-Aldrich | C2206 | 1/100 for IF |
| Antibody | anti-β-catenin mouse monoclonal | BD Biosciences | 610156 | 1/100 for IF |
| Antibody | recombinant anti-paxillin rabbit monoclonal antibody | Abcam | Ab32084 | 1/100 for IF |
| Antibody | mouse anti-GAPDH | ProteinTech | 60004–1-Ig | 1/100 for IF |
| Antibody | mouse anti-Arp3 | Sigma-Aldrich | A5979 | 1/100 for IF |
| Antibody | mouse anti-E-cadherin | BD Biosciences | 610181 | 1/100 for IF |
| Antibody | rabbit anti-α-catenin polyclonal | Sigma-Aldrich | C-2081 | 1/100 for IF |
| Antibody | rat anti-α18-catenin monoclonal | generously provided by A. Nagafuchi, (Kumamoto University, Japan) | | 1/100 for IF |

*Continued on next page*

*Continued*

| Reagent type (species) or resource | Designation | Source or reference | Identifiers | Additional information |
|---|---|---|---|---|
| Antibody | Alexa488- | Life Technologies | A11039, A11055, A11013 | 1/250 for IF |
| Antibody | Alexa568- | Life Technologies | A11004, A11011, A11077 | 1/250 for IF |
| Antibody | Alexa647- | Life Technologies | A31571, A31573 | 1/250 for IF |
| Chemical compound, drug | Alexa (488) - coupled phalloidins | Invitrogen | A12379 | 1/250 for IF |
| Chemical compound, drug | Alexa (555 or 647) - coupled phalloidins | Life Technologies | A34055, A22287 | 1/250 for IF |
| Other | Hoechst 34580 | ThermoFisher | H3570 | 1/10000 for IF |
| Antibody | Horseradish peroxidase-coupled anti-mouse IgGs | Sigma-Aldrich | A9044 | 1/10000 for WB |
| Antibody | Horseradish peroxidase-coupled anti-rabbit IgGs | Pierce | | 1/10000 for WB |
| Chemical compound, drug | Mitomycin C | Sigma-Aldrich | M2487 | 10 µg/ml for 1 hr |
| Chemical compound, drug | Y-27632 dihydro chloride | Sigma-Aldrich | Y0503 | 50 µM |
| Other | APP (Azido-Poly-lysine Poly (ethylene glycol)) | Inspired protocol from M. van Dongen, Matthieu Piel | https://doi.org/10.1002/adma.201204474 | Inspired protocol from M. van Dongen, Matthieu Piel |
| Peptide, recombinant protein | BCN-RGD peptide (BCN: bicyclo[6.1.0]-nonyne, coupled to RGD: peptide sequence Arg-Gly-Asp) | Inspired protocol from M. van Dongen, Matthieu Piel | https://doi.org/10.1002/adma.201204474 | Inspired protocol from M. van Dongen, Matthieu Piel |
| Commercial assay or kit | DMEM (containing Glutamax, High Glucose and Pyruvate) | Life Technologies | 31966–021 | |
| Commercial assay or kit | Fluorobrite DMEM | Thermo Fisher | A18967-01 | |
| Commercial assay or kit | Penicillin/ Streptomycin | Life Technologies | 15140–122 | |
| Commercial assay or kit | Foetal Bovine Serum | Life Technologies | S1810-500 | 10% FBS in DMEM |
| Commercial assay or kit | geneticin | Life Technologies | 10131–019 | |
| Chemical compound, drug | Trypsin | Life Technologies | 25300–054 | |
| Genetic reagent (Plasmid) | pLKO.1-puro | Sigma-Aldrich | SHC002 | |
| Genetic reagent (Plasmid) | MYH9 | Sigma-Aldrich | transcript ID: ENSCAFT00000002643.3 | TTGGAGCCATA CAACAAATAC for NMIIA |
| Genetic reagent (Plasmid) | MYH10 | Sigma-Aldrich | transcript ID: ENSCAFT00000027478 | TCGGGCAGCTCTA CAAAGAAT for NMIIB |

*Continued on next page*

*Continued*

| Reagent type (species) or resource | Designation | Source or reference | Identifiers | Additional information |
|---|---|---|---|---|
| Genetic reagent (Plasmid) | RFP-Pericentrin | kindly provided M. Coppey, Institut Jacques Monod, Paris | | kindly provided M. Coppey, Institut J acques Monod, Paris |
| Genetic reagent (Plasmid) | m-Cherry cortactin | kindly provided by Alexis Gautreau, Biochemisty laboratory, Ecole polytec hnique, France | https://portail.po lytechnique.edu/bioc/ en/gautreau | pcDNA5-FRT-GFP-mCherry-3pGW back bone (1740-pcDNAM FRTPC-mCherry Cortactine) |
| Genetic reagent (Plasmid) | mCherry Myosin IIB | Addgene | 55107 | |
| Genetic reagent (Plasmid) | CMV-GFP-NMHC II-A | Addgene | 11347 | |
| Chemical compound, drug | protease inhibitor cocktail | Roche | 27368400 | |
| Chemical compound, drug | phosphatase inhibitor (Phosphostop) | Roche | 4906837001 | |
| Commercial assay or kit | Bradford assay | BioRad | 500–0006 | |
| Commercial assay or kit | 4–12% Bis-Tris gel | Novex | NP0335 | |
| Commercial assay or kit | Supersignal west femto maximum sensitivity substrate | ThermoFisher | 34095 | |
| Commercial assay or kit | LookOut Mycoplasma PCR detection Kit | Sigma-Aldrich | MP0035 | |
| Chemical compound, drug | paraform aldehyde | Thermo Scientific | 22980 | |
| Chemical compound, drug | Fluoromount-G mounting media | Southern Biotech | | |
| Peptide, recombinant protein | fibronectin | Merck Millipore | FC010 | |
| Chemical compound, drug | APTES | Sigma-Aldrich | A3648 | |
| Chemical compound, drug | EDC-HCl | Thermo Scientific | 22980 | 2 mM freshly prepared in 0.1M MES pH4.7 |
| Chemical compound, drug | NHS | Sigma-Aldrich | 130672 | 5 mM |
| Peptide, recombinant protein | recombinant human E-cadherin | R and D systems | 8505-EC | 1 µg |
| Chemical compound, drug | Cy 52–276 A and Cy 52–276 B silicone elastomer | Dow corning | | |
| Chemical compound, drug | carboxylated red fluorescent beads | Invitrogen | F8801 | |

*Continued on next page*

*Continued*

| Reagent type (species) or resource | Designation | Source or reference | Identifiers | Additional information |
|---|---|---|---|---|
| Software, algorithm | FIJI-Image J | https://imagej.net/Fiji/Downloads | Image analysis were done using Fiji-Image J and plugins | |
| Software, algorithm | MATLAb | MATLAB | Traction force, PIV analysis were done using alogorithms developed in lab to analyse traction force | |
| Software, algorithm | Photoshop and Illustrator | Adobe | Images were mounted using these softwares | |
| Software, algorithm | GraphPad prism | GraphPad Prism | Graphs and statistical tests were done using GraphPad Prism | |

## Antibodies and reagents

The following primary antibodies were used: rabbit anti-NMIIA polyclonal (Biolegend) or mouse anti-NMIIA monoclonal antibodies (Abcam, for co-immunostainings with anti-NMIIB antibodies); rabbit anti-β-catenin polyclonal (Sigma-Aldrich) or mouse anti-β-catenin monoclonal (BD Biosciences) antibodies; recombinant rabbit anti-paxillin monoclonal antibody (Abcam); mouse anti-GAPDH (Protein-Tech), mouse anti-Arp3 (Sigma-Aldrich) and mouse anti-E-cadherin (BD Biosciences) antibodies; rabbit anti-α-catenin polyclonal (Sigma-Aldrich) and rabbit anti-NMIIB polyclonal (Biolegend) antibodies; rat anti-α18-catenin monoclonal antibody (generously provided by A. Nagafuchi (Kumamoto University, Japan) (*Yonemura et al., 2010*). Alexa488-, Alexa568- and Alexa647-conjugated secondary antibodies were purchased from ThermoFisher, Alexa (488 or 555 or 647) -coupled phalloidins from Invitrogen and Hoechst 34580 from ThermoFisher. Horseradish peroxidase-coupled anti-mouse IgGs (Sigma-Aldrich) and anti-rabbit IgGs (Pierce) were used for immunoblotting. Mitomycin C and Y-27632 dihydrochloride were purchased from Sigma-Aldrich. The APP (Azido-Poly-lysine Poly (ethylene glycol)) and the BCN-RGD peptide (BCN: bicyclo[6.1.0]- nonyne, coupled to RGD: peptide sequence Arg-Gly-Asp) were prepared as previously described (*van Dongen et al., 2013*).

## Cell culture

MDCK (ATCC CCL-34) and C2C12 (ATCC CRL-1772) cells originate from the American Type Culture Collection (ATCC). E-cadherin-GFP (*Adams et al., 1998*) and α-catenin KD MDCK cell lines (*Benjamin et al., 2010*) were kindly provided by W.J. Nelson (Stanford University, Palo Alto). Caco2BBE cells (ATCC HTB-37) were kindly provided by S. Robine (Institut Curie/CNRS, Paris). Cells were maintained at 37°C, 5% CO2 in DMEM (containing Glutamax, High Glucose and Pyruvate, Life Technologies) supplemented with 100 μg/mL Penicillin/Streptomycin (Life Technologies) and Foetal Bovine Serum (Life Technologies) at 10% for MDCK and C2C12 cells and at 20% for Caco2 cells. Ecadherin-GFP cells and α-catenin KD MDCK cells were maintained in media containing 5 μg/ml geneticin (Life Technologies).

## Generation of isoform-specific NMII knock-down MDCK cell lines

For generation of isoform-specific NMII Heavy chain knock-down cells, isoform-specific shRNA sequences, inserted in a back bone standard vector pLKO.1-puro, were designed and synthetized by Sigma-Aldrich technical services, based on the sequences of Canis lupus familiaris transcripts for MYH9 (NMIIA, transcript ID: ENSCAFT00000002643.3) and MYH10 (NMIIB, transcript ID: ENSCAFT00000027478). The sequences used were the following: TTGGAGCCATACAACAAATAC for NMIIA and TCGGGCAGCTCTACAAAGAAT for NMIIB. As a control, the pLKO.1-puro non-

mammalian shRNA Control Plasmid DNA was used (SHC002, Sigma-Aldrich). Two million Ecadherin-GFP MDCK cells were electroporated (Neon Transfection System Invitrogen) with 3–5 ug shRNA encoding plasmids in one pulse of 20 ms at 1650 V. Twenty four hours later, cells were put under selection pressure by adding puromycin (2.5 µg/ml) in media. After 10 days, single cells were sorted in 96 well plates by flow cytometry using Influx 500 sorter-analyzer (BD BioSciences) and clonal populations then selected based on NMII isoform expression levels by immunoblot and immunofluorescence. Control, NMIIA KD and NMIIB KD MDCK cells were maintained in media containing geneticin (5 µg/ml) and puromycin (2.5 µg/ml), cells were tested and verified for absence of mycoplasma using LookOut Mycoplasma PCR detection kit (sigma-aldrich MP0035).

For simultaneous visualization of E-cadherin and centrosome, Ecadherin-GFP MDCK cells were transiently transfected with a plasmid driving the expression of RFP-Pericentrin (kindly provided M. Coppey, Institut Jacques Monod, Paris), using the protocol described above, one or two days before the experiment. m-Cherry cortactin plasmids (kindly provided by Alexis Gautreau, Biochemisty laboratory, Ecole polytechnique, France) were transfected in Control, NMIIA KD and NMIIB KD MDCK cells and the m-cherry expressing cell population was sorted by flow cytometry using Influx 500 sorter-analyzer (BD BioSciences). For expression of exogenous NMIIA and NMIIB in MDCK cells, the WT MDCK cells were transiently transfected with a plasmid driving the expression of GFP-NMIIA (Addgene 11347) or m-cherry NMIIB (Addgene 55107), using the protocol described above.

## Western blotting

Confluent cells were lysed in 100 mM Tris pH 7.5,150 mM NaCl, 0.5% NP40, 0.5% triton-X100, 10% glycerol,1X protease inhibitor cocktail (Roche) and 1X phosphatase inhibitor (Phosphostop, Roche) for 20 min at 4°C. Insoluble debris were centrifuged for 15 min at 13000 g and supernatants were recovered. Protein concentration was quantified by Bradford assay (BioRad), SDS PAGE and electrotransfer were performed on 4–12% Bis-Tris gel (Novex) using mini gel tank and iBlot transfer systems (Invitrogen). Non-specific sites were blocked with 5% non-fat dry milk in PBS 0.1% Tween 20. Primary antibodies were diluted (1/1000) in PBS 0.1% Tween 20 and incubated overnight at 4°C. After three washes in PBS 0.1% Tween 20, secondary HRP antibodies diluted in PBS 0.1% Tween 20 (1/10000) were incubated for 1 hr and washed 3 times with PBS 0.1% Tween 20. Immunocomplexes of interest were detected using Supersignal west femto maximum sensitivity substrate (ThermoFisher) and visualized with ChemiDoc chemoluminescence detection system (Biorad). Quantification of Western blots by densitometry was performed using the Gel analyzer plug in from Image J. GADPH was used as a loading control to normalize the quantification.

## Immunofluorescent staining

Cells were fixed with pre-warmed 4% formaldehyde in PBS for 15 min at RT and then washed 3 times with PBS, followed by permeabilization and blocking with 0.05% saponin/0.2% BSA in PBS for 15 min at RT. The primary antibodies diluted (1/100) in Saponin/BSA buffer were then incubated overnight at 4°C. After three washes in Saponin/BSA buffer, the samples were incubated with secondary antibodies (1/250) and Alexa-coupled phalloidin, diluted at 1/200 in the same buffer for 1 hr at RT. The preparations were washed twice in Saponin/BSA buffer, once in PBS, and then mounted with the DAPI Fluoromount-G mounting media (Southern Biotech).

## Preparation of fibronectin-coated and cadherin-coated substrates

For fibronectin coating, glass coverslips were first cleaned by sonication in 70% ethanol and air dried. They were coated for 1 hr with 50 µg/mL human plasma fibronectin (Merck Millipore) diluted in PBS and washed three times with PBS.

The protocol for E-cadherin coating was inspired from a previous study by Lee and colleagues (*Lee et al., 2016*). Briefly, the cleaned glass coverslips were silanized with 10% 3-aminopropyl triethoxysilane (APTES, Sigma-Aldrich) in 100% ethanol for 10 min at RT, washed once in 100% ethanol and dried at 80°C for 10 min. The surface was then functionalized by incubation for 1 hr with 2 mM EDC-HCl (Thermo Scientific)/5 mM NHS (Sigma-Aldrich) and 1 µg of recombinant human E-cadherin (R and D systems). Coverslips were then washed two times with PBS.

Cells were plated at very low density (typically 1 105 cells for a 32 mm diameter coverslip) on the coated coverslips in complete medium containing 10 µg/mL mitomycin C. After 1 hr incubation at

37°C, the preparations were washed twice with complete media and incubated 2–6 hr or overnight at 37°C before imaging or fixation, for cadherin coating and fibronectin coating, respectively.

## Preparation of switchable micro-patterns and imaging

Micropatterns were made as previously described with some modifications (*van Dongen et al., 2013*). Briefly, air dried cleaned glass coverslips were activated with deep UV for 5 min, and coated for at least 1 hr with the repellent compound APP (0.1 mg/ml in HEPES 10 mM pH7.4). After three washes with deionized water, the coverslips were exposed to deep UV for 7 min through a chrome photomask. The coverslips were then washed with deionized water three times, coated with 50 μg/mL human plasma fibronectin for 1 hr and washed twice with deionized water and once with PBS. When indicated, the coating was done with a 2:1 ratio of non-coupled:Cy3-coupled fibronectin prepared with Cy3 Mono-Reactive Dye Pack (GE Healthcare) as recommended by the manufacturer.

Cells were resuspended at $4.10^2$ cells/mm$^2$ in medium containing 10 μg/mL mitomycin C and deposited on the patterned slide. After 1 hr of incubation at 37°C, cells were washed 3 times with fresh medium to remove mitomycin C and cells that remained in suspension. The cells that adhered on micro-patterns were left overnight in the incubator. The day after, confinement was released by addition of 20 μM BCN-RGD peptide diluted in DMEM media or, in case of live-imaging experiments, in Fluorobrite DMEM (Thermo Fisher) supplemented with 10% FBS and 1% Penicillin/Streptomycin. For ROCK inhibition experiments, 50 μM Y-27632 was added at the same time as BCN-RGD. Samples were then immediately imaged under a microscope or left in the incubator for 20 more hours and fixed as described above. When indicated for live-imaging experiments, nuclei were stained before adding BCN-RGD peptide by incubating the preparations with 5 μg/mL Hoechst 34580 in the medium for 20 min at 37°C followed by two washes with fresh media.

## Image acquisition and analysis

For live-microscopy experiments, the samples were placed in a chamber equilibrated at 37°C under 5% CO2 atmosphere. Images were acquired with a Yokogawa-Andor CSU-W1 Spinning Disk confocal mounted on an inversed motorized Leica DMI8 microscope and equipped with a sCMOS Orca-Flash 4 V2+ camera (Hamamatsu) and a 63 X oil immersion objective or a 20 X dry objective, with multi-positioning and a resolution of 0.5–3 μm z-stacks. Alternatively, the samples were imaged with an Olympus IX81 wide-field fluorescence microscope equipped with a Coolsnap HQ CCD camera and a 60X oil immersion objective or a 20 X dry objective. For some experiments, the Nikon Biostation IM-Q microscope was also used with 10X or 20X objective and multi-positioning.

For fixed samples, images were acquired with a Zeiss Apotome fluorescence microscope equipped with a 63 X oil immersion objective or with a Zeiss LSM 780 confocal microscope equipped with a 63 X oil immersion objective at a resolution of 0.3 μm z-stacks.

Image processing and analysis were done on Fiji software. Analysis of junction parameters (length, straightness and angle deviation) was done manually with Fiji software based on phase contrast and GFP-Ecadherin signal. Cell spreading, focal adhesions and α-catenin clustering were analyzed by thresholding the image and applying an 'Analyze particles' which gives the number of objects and its area. To calculate the ratio of α-cat to α18-cat intensities, the mean gray intensity value for the two channels were measured within the manually-defined junction. Tracking of single cells on fibronectin was done using the Manual Tracking plugin. For colocalization analysis, Pearson's correlation coefficient was calculated using the Coloc2 Plugin from image J, on an ROI corresponding to the junction area. For relative intensiy profiles, a line was drawn on the ROI and the line scan was done using the plot profile plugin in image J, the values obtained were then normalized to the maximal intensity of each channel.

## Traction force microscopy

Soft silicone elastomer substrates for TFM (Traction force microscopy) were prepared as described previously with some modifications (*Vedula et al., 2014*). Cy 52–276 A and Cy 52–276 B silicone elastomer components (Dow corning) were mixed in a 5:5 (elastic modulus ~15 kPa for E-cadherin-coating) or a 5:6 ratio (elastic modulus ~30 kPa for fibronectin-coating). 0.08 g of elastomer was deposited on 32 mm glass coverslips and allowed to spread progressively. The substrate was silanized with 10% (3-aminopropyl triethoxysilane (APTES, Sigma) in 100% ethanol for 10 min at RT,

washed once in 100% ethanol and dried at 80°C for 10 min. The surface was coated for 10 min at RT with carboxylated red fluorescent beads (100 nm, Invitrogen) diluted at 2-3/1000 in deionized water. After washing with deionized water, the surface was finally functionalized with protein (fibronectin or E-cadherin) as described above. Seeded cells together with fluorescent beads were imaged either on an Olympus-CSU-W1 Spinning Disk confocal microscope with a 10 X dry objective and 3 μm z stacks or on an Olympus-IX81 wide field inverted fluorescence microscope with a 20 X dry objective for 2 to 24 hr, at a frequency of 1 frame every 10 min, at 37°C under 5% CO2 atmosphere. At the end of the acquisition, 100–200 μL of 10% SDS was added in the media to detach cells and image a reference frame. For force calculation, matPIV was used to analyse the displacement vectors of the beads, which were further translated into forces using the FTTC plugin in ImageJ. The vector quiver plots and heat map of magnitude force was plotted using Matlab. Mean (resp. resultant) forces exerted by cells and doublets were obtained by computing the average of the magnitude (resp. the vectorial sum) of traction forces within manually defined masks. For the analysis of tractions forces below cell-cell junctions, the junction masks and corresponding midline were first manually defined based on the E-cadherin-GFP pictures. Then, the midline was used to define the average orientation of the junction, and all force vectors within the junction mask were projected onto the directions parallel and perpendicular to this orientation. The mask was divided in four quarters along this mean orientation. The 'junction centre parallel (resp. perpendicular) force' is defined as the averaged absolute value of the parallel (resp. perpendicular) component of traction forces in the two central quarters of the mask, while the 'junction periphery parallel (resp. perpendicular) force' is the averaged absolute value of the parallel (resp. perpendicular) component of traction forces in the two outermost quarters.

$$T_{parallel/perpendicular}(center/periphery) = \langle |T_{parallel/perpendicular}| \rangle_{center/periphery}$$

## Calculation of inter-cellular stress

Computing the junctional stress components $\sigma_=$ and $\sigma_\perp$, respectively parallel and perpendicular to the cell junction (*Figure 6d*), required both the determination of the cell junction location and the estimation of the inter-cellular stress tensor. The cell junction domain was defined as the overlap between two masks representing the area covered by each cell in the doublet. Given the stress tensor, the parallel and perpendicular stress components were obtained by rotation from the cartesian basis. As exemplified in *Figure 6a*, we found in most cases that the cell junction domain was roughly straight: the mean orientation of the cell junction domain determined the rotation angle. We checked that following the cell junction contour did not significantly modify our estimates. Finally, each junctional stress component was spatially-averaged over the cell junction domain.

Intercellular stress was estimated by Bayesian inversion (*Nier et al., 2016*), with a dimensionless regularization parameter $\Lambda = 10^5$ (see *Harris et al., 2014* for details). The spatial domain for stress estimation was for each image the smallest rectangle encompassing the cell doublet. For simplicity, we implemented free stress boundary conditions on the straight boundaries of the rectangular domain, instead of following the cell doublet boundaries. As a consequence, the stress estimation was qualitative, but sufficed to evaluate differences between conditions. Note that height variations within the cell doublet were also neglected in the estimation of the 2D inter-cellular stress field.

## SIM microscopy

Super-resolution structured-illumination microscopy was performed on a Zeiss Elyra PS.1 microscope with a 63 X objective (Plan Apo 1.4NA oil immersion) and an additional optovar lens 1.6 X. Cells grown on 0.17 mm high-performance Zeiss coverslips were fixed and prepared for immunostaining, then with DAPI Fluoromount-G mounting media (Southern Biotech). Laser lines 488 nm, 561 nm and 641 nm were directed into the microscope, passing through a diffraction grating. For 3D SIM imaging, the diffraction grating was rotated along three directions (angles 120o) and translated (five lateral positions) throughout the acquisition. Typically, 20–30 slices of 110 nm were acquired for each cell corresponding to an imaging height of 2–3 μm. The fluorescence signal was detected with an EMCCD camera (iXon-885, Andor, 1004 × 1002, pixel size 8 μm, QE = 65%). Processed SIM images were aligned via an affine transformation matrix of predefined values obtained using 100 nm multi-color Tetraspeck fluorescent microspheres (Thermo Fisher Scientific).

## Data display and statistics

Images were mounted using Photoshop and Illustrator. Graphs and statistical tests were done using GraphPad prism software.

## Acknowledgements

This work was supported by the European Research Council (Grant No. CoG-617233), LABEX 'Who Am I?' (ANR-11-LABX-0071), CNRS "Défi Mechanobiology" AAP 2016, the Ligue Contre le Cancer (Equipe labellisée) and the Agence Nationale de la Recherche 'POLCAM' (Grant No. ANR-17-CE13-0013). We acknowledge the ImagoSeine core facility of the Institut Jacques Monod, member of IBiSA and France-BioImaging (ANR-10-INBS-04) infrastructures. We thank Orestis (ImagoSeine core facility, IJM) for technical assistance with SIM experiments. We thank Sree Vaishnavi and Gianluca Grenci (Micro fabrication Core Facility of Mechnabiology Institute, National University of Singapore) for the fabrication of chrome photomask. We thank A Nagafuchi for α18-catenin antibody, WJ Nelson and S Robine for providing cells, M Coppey for RFP-Pericentrin plasmid and M Piel (IPGG, Curie Institute) for providing original APP and BCN-RGD compounds. We thank Delphine Delacour and Shreyansh Jain for useful scientific discussions.

## Additional information

### Funding

| Funder | Grant reference number | Author |
| --- | --- | --- |
| Seventh Framework Programme | CoG-617233 | Benoit Ladoux |
| Agence Nationale de la Recherche | ANR-17-CE13-0013 | René-Marc Mège |
| Agence Nationale de la Recherche | ANR-10-INBS-04 | René-Marc Mège Benoit Ladoux |
| Agence Nationale de la Recherche | ANR-11-LABX-0071 | René-Marc Mège Benoit Ladoux |
| Agence Nationale de la Recherche | ANR-11-LABX-0071 | Benoit Ladoux |
| Agence Nationale de la Recherche | ANR-17-CE13-0012 | Benoit Ladoux |
| Ligue Contre le Cancer | Equipe Labellisée | René-Marc Mège |
| Centre National de la Recherche Scientifique | Défi Mechanobiology | René-Marc Mège |

The funders had no role in study design, data collection and interpretation, or the decision to submit the work for publication.

### Author contributions

Mélina L Heuzé, Conceptualization, Data curation, Software, Formal analysis, Validation, Investigation, Visualization, Methodology, Writing—original draft, Writing—review and editing; Gautham Hari Narayana Sankara Narayana, Data curation, Formal analysis, Validation, Investigation, Visualization, Methodology, Writing—original draft; Joseph D'Alessandro, Resources, Software, Investigation, Methodology; Victor Cellerin, Formal analysis, Methodology, Writing—original draft; Tien Dang, Formal analysis, Methodology; David S Williams, Software, Formal analysis, Methodology; Jan CM Van Hest, Formal analysis, Methodology, Software; Philippe Marcq, Conceptualization, Resources, Software, Formal analysis, Supervision, Funding acquisition, Methodology, Writing—original draft, Project administration, Writing—review and editing; René-Marc Mège, Benoit Ladoux, Conceptualization, Resources, Formal analysis, Supervision, Funding acquisition, Methodology, Writing—original draft, Project administration, Writing—review and editing

## Author ORCIDs

Mélina L Heuzé (iD) https://orcid.org/0000-0002-4271-2706
Gautham Hari Narayana Sankara Narayana (iD) https://orcid.org/0000-0002-2534-5836
Joseph D'Alessandro (iD) http://orcid.org/0000-0002-1585-3255
René-Marc Mège (iD) https://orcid.org/0000-0001-8128-5543
Benoit Ladoux (iD) https://orcid.org/0000-0003-2086-1556

## Decision letter and Author response

Decision letter https://doi.org/10.7554/eLife.46599.035
Author response https://doi.org/10.7554/eLife.46599.036

## Additional files

### Supplementary files
• Transparent reporting form
DOI: https://doi.org/10.7554/eLife.46599.032

### Data availability
All data generated or analysed during this study are included in the manuscript and supporting files. Source data files have been provided for the main figures and figure supplements.

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
