## [Decision Letter]

Thank you for submitting your article "Myosin II isoforms play distinct roles in adherens junction biogenesis" for consideration by *eLife*. Your article has been reviewed by three peer reviewers, and the evaluation has been overseen by a Reviewing Editor and Anna Akhmanova as the Senior Editor. The following individual involved in review of your submission has agreed to reveal their identity: Dennis Discher (Reviewer #2).

The reviewers have discussed the reviews with one another and the Reviewing Editor has drafted this decision to help you prepare a revised submission.

Summary:

This paper explores the roles of non-muscle myosin isoforms in cell-cell junction formation. Similar studies using siRNA knockdown or CRISPR knockout have been reported, but the direct assessment of the roles of NMIIA and B on tension anisotropy, as well as exploration of the roles of these isoforms on developing junctions, are significant contributions. Some of the results presented here contradict earlier studies, and the paper needs to carefully compare and discuss the present results to these earlier studies. In addition, there are several concerns that must be addressed in a revised manuscript.

Essential revisions:

1) Figures 3 and 4 show the localization of NMIIA and NMIIB at and near adherens junctions with regular confocal microscopy (Figure 3) and structured illumination microscopy (Figure 4). The two figures are inconsistent in that Figure 3 shows NMIIB only associated with the E-cad/β-catenin/membrane and absent from the perijunctional F-actin cables, whereas the SIM images clearly show NMIIB localized to both structures. The authors appear to go along with the results from Figure 3 and ignore the clear co-localization of NMIIB with NMIIA in F-actin cables on both sides of the cell-cell junction in their Results and Discussion. Given that much of the authors' model rests on results arising from the use of immunocytochemistry (which depends on the robustness of the two antibodies used here) to differentially localize myosin IIA and IIB, it is important to confirm them using either knock-in GFP labels, or else exogenous myosin constructs expressed at levels comparable to that of the native proteins. Do the SIM images in particular reproduce with GFP-NMIIB expressed at endogenous levels? These issues are especially important as the authors provide data that are qualitatively different from the Yap lab papers. Finally, quantification of the data in Figures 3-5 should be provided.

2) From the very widely distributed data in Figure 8D the authors conclude that in contrast to the anisotropic force distribution at junction of control cells NMIIB KD cells exhibit isotropic force distribution. This is based on the mean values for a very wide distribution. It would be more convincing if they compared each pair of measurements (perpendicular and parallel force) from the same junction and did the pairwise statistics. Also, it was not possible to discern tension and compression in Figure 8A as indicated in the text and the illustration to the right of the panels. Something appears to be missing here.

3) It is essential that the manuscript's results be placed in the context of prior studies. The authors should cite and discuss a paper by Ozawa 2018, which explicitly examines the contribution of myosin IIA to junctional stabilization in MDCK cells. In addition, the work by Smutny et al. (which is cited) shows that the formation of cadherin clusters on surfaces requires myosin IIA, not myosin IIB, in apparent contradiction with the present study. The authors should acknowledge and discuss this puzzling difference. More broadly, Smutny et al. conclude that "[…] myosin IIB supports integrity of the apical actin ring to prevent fragmentation of the ZA, perhaps by reinforcing it to resist disruptive orthogonal forces." This presages a major outcome of the present study, and should be acknowledged. Finally, Gomez et al. examine the specific localization of myosin IIA and IIB in the context of epithelial cell-cell junctions in MCF7 cells (Gomez et al., 2015), and do not appear to observe the differential localization that the authors here report. This difference should likewise be acknowledged. Overall these differences, which could reflect the use of different cell lines, strongly imply that more caution should be employed in interpreting the present study

4) The authors claim that "NMIIB supports junctional branched actin organization" (Figure 5) is problematic in several ways. First of all, they provide no evidence that the cortical actin network associated with NMIIB is an arp2/3-dependent branched actin network. It is true that other publications showed arp2/3 localizes to junctions, but so do formins. The cortex is probably nucleated by a combination of nucleators, so the authors shouldn't refer to this juxtamembrane network as branched without any evidence. Second, in Figure 5A or elsewhere I don't see any evidence of a disappearing actin network between the control and NMIIB KD images. So, while it is true that NMIIB associates with membrane proximal actin there are no experimental data that support this claim. This claim needs to be discussed more carefully or dropped.

5) Figure 6 shows that single NMIIA KD cells barely generate any traction forces on ECM. Figure 8 shows the same with cell doublets. It isn't clear what is learned about the role of NMIIA in generating tension at cell-cell junctions from this experiment. In contrast, Figure 6 shows that the traction forces of NMIIB KD cells on ECM are the same as control. In doublets, Figure 8 shows that the overall traction forces are nearly the same as control, but they observe forces underneath the cell-cell junctions – something not observed in control. This suggests that the force transmission across the cell-cell junction is perturbed and that is why forces are felt at the cell-ECM interface. This should be considered as an explanation.

[Editors' note: further revisions were requested prior to acceptance, as described below.]

Thank you for resubmitting your work entitled "Myosin II isoforms play distinct roles in adherens junction biogenesis" for further consideration at *eLife*. Your revised article has been favorably evaluated by Anna Akhmanova as the Senior Editor, a Reviewing Editor, and two reviewers.

The manuscript has been improved but there are some remaining issues that need to be addressed before acceptance, as outlined below:

The reviewers agree that you have addressed most of the points raised in the original submission. However, there are still concerns regarding overstated claims of novelty and perhaps interpretations of published data. While you have been careful in the Discussion to emphasize that you are looking at early stages of junction formation, which is an important and new aspect of your work, some of the data concerns junctions in fully polarized cells, which do overlap the work of Yap and Ozawa. Thus, before this manuscript can be accepted, several points need to be addressed in order to place your findings in the proper context of previous papers. None of these points require new experiments, but rather small modifications to the text. Please consider these carefully and modify accordingly.

1) The conclusion of the Introduction makes it sounds like this topic has not been investigated previously. A simple fix would be to change "questioned the unexplored functions […]" to "further explore the functions […]" or something similar.

2) In the subsection “NMIIA and NMIIB orchestrate junction biogenesis”, you need to add a sentence acknowledging that Smutny 2010 reported similar findings in mature junctions. Likewise, the phrase "explored the possibility" is misleading given that Smutny had found NMII isoform differences in regulating junctional actin, and this should be acknowledged.

3) There are some differences in the differential localization of the NMII isoforms with the Smutny 2010 and Gomez 2015 papers. It appears that some of this is due to your careful examination of NMIIb colocalized with cadherin, whereas the other papers seem to have looked only at actin. It would be helpful to consider this difference.

4) At first glance your results with α-catenin would seem to contradict those of the Ozawa 2018 paper. However, since you report a lowering of total α-catenin at junctions (subsection “NMIIA regulates the organization of perijunctional actin bundles while NMIIB regulates the organization of a juxtamembrane actin layer”, last paragraph), there is likely no real discrepancy. However, you should discuss this in the Discussion.

---

## [Author Response]

Essential revisions:1) Figures 3 and 4 show the localization of NMIIA and NMIIB at and near adherens junctions with regular confocal microscopy (Figure 3) and structured illumination microscopy (Figure 4). The two figures are inconsistent in that Figure 3 shows NMIIB only associated with the E-cad/β-catenin/membrane and absent from the perijunctional F-actin cables, whereas the SIM images clearly show NMIIB localized to both structures. The authors appear to go along with the results from Figure 3 and ignore the clear co-localization of NMIIB with NMIIA in F-actin cables on both sides of the cell-cell junction in their Results and Discussion.

We thank the reviewers for raising this somehow unclear statement in our initial statements. Indeed, as noticed by the reviewers on SIM images (Figure 4), NMIIB is strongly accumulated at cell-cell contacts at the membrane, but it is also detectable on thick perijunctional F-actin cables, albeit with a lower intensity (see line scans Figure 4C and Figure 3—figure supplement 2D). There is however no inconsistency between Figure 3 and Figure 4, as the presence of perijunctional NMIIB is also seen by confocal microscopy (see Figure 3G Day1 and Figure 3—figure supplement 1B and Figure 3—figure supplement 2C). This is however quite variable from cell to cell. Another example of NMIIB distribution at cell-cell contacts, imaged by confocal microscopy, is provided as Author response image 1. Thus, the reviewers are right with the fact that our statement on the distribution of NMIIB was mostly centred on membrane-associated localization. We now scrutinized the whole manuscript to correct this and more accurately describe and interpret NMIIB localization at cell-cell contacts (modifications spread over the subsection “NMIIB preferentially localizes to a junctional actin pool distinct from perijunctional NMIIA-associated contractile fibres” and subsection “NMIIA regulates the organization of perijunctional actin bundles while NMIIB regulates the organization of a juxtamembrane actin layer”). This modulation of our message however does not jeopardize the conclusions of our work.

**Author response image 1. respfig1:** NMIIB localizes to early AJs. (**a**) Representative confocal images and zoom boxes of GFP-E-cadherin-expressing MDCK cell doublets fixed 20h after BCN-RGD addition and immuno-stained for NMIIB. Scale bar: 10 μm. (**b**) Relative intensity profiles (raw and smoothed data) of GFP-E-cadherin and NMIIB signals along the line represented in (**a**). Blue arrows point at NMIIB that sits on parallel perijunctional F-actin cables.

Given that much of the authors' model rests on results arising from the use of immunocytochemistry (which depends on the robustness of the two antibodies used here) to differentially localize myosin IIA and IIB, it is important to confirm them using either knock-in GFP labels, or else exogenous myosin constructs expressed at levels comparable to that of the native proteins. Do the SIM images in particular reproduce with GFP-NMIIB expressed at endogenous levels?

We followed the advice of the reviewers analysing the distribution of exogenously expressed mCherryNMIIB and GFP-NMIIA. Although we could not compare the expression to the one of endogenous proteins in short term transfections, we confirm that mCherry-NMIIB predominantly localized at junctional membranes, and to a lesser extent on perijunctional F-actin cables (see line scan Figure 3—figure supplement 1E) while GFP-NMIIA predominantly associated with contractile actin bundles. This is illustrated in Figure 3—figure supplement 1D, E.

These issues are especially important as the authors provide data that are qualitatively different from the Yap lab papers.

In fact, our data are divergent from the Smutny et al. 2010 paper but fit perfectly well with the report of the same lab in 2015 (Gomez et al., 2015) and as well with the Ozawa et al. 2018 paper. Our statements in our first draft were somehow unclear and were not referring to the most recent reports (Gomez 2015 and Ozawa 2018). Our data are fully consistent with these two papers reporting a junctional localization of NMIIB and a perijunctional localization of NMIIA. Our data are now discussed at the light of these three papers (subsection “NMIIB preferentially localizes to a junctional actin pool distinct from perijunctional NMIIA-associated contractile fibres”).

Finally, quantification of the data in Figures 3-5 should be provided.

Quantitative analysis of the colocalization of NMIIA or NMIIB toward E-cadherin stainings has been performed for Figure 3A, B, F, G, Figure 4A, B and Figure 5B). We choose for these quantifications to apply the Pearson’s coefficient. Results can be seen in Figure 3E, Figure 4E, Figure 5D, respectively. Additional line scans are shown in Figure 4C, Figure 5C, F and Figure 3—figure supplement 2D, F.

2) From the very widely distributed data in Figure 8D the authors conclude that in contrast to the anisotropic force distribution at junction of control cells NMIIB KD cells exhibit isotropic force distribution. This is based on the mean values for a very wide distribution. It would be more convincing if they compared each pair of measurements (perpendicular and parallel force) from the same junction and did the pairwise statistics.

We thank the reviewers for raising this point. As advised, we have now performed pairwise statistical t tests in Figure 6D to compare parallel and perpendicular stress from the same junction at the same time point in Ctrl, NMIIA KD and NMIIB KD doublets. The results remain unchanged, showing a preferentially parallel anisotropic stress in Ctrl junctions while the stress appears isotropic in NMIIA KD and NMIIB KD junctions. Pairwise statistical t tests were also applied in Figure 6—figure supplement 3B to compare parallel and perpendicular forces at the center and periphery of the junction in Ctrl and NMIIB KD doublets. Moreover, data from additional experiments were added to all the panels of TFM experiments on fibronectin (Figure 6, Figure 6—figure supplement 1D and Figure 6—figure supplement 3). These new data did not modify the results and even decreased, in some conditions, the p value obtained initially with statistical tests, which makes our results even more convincing.

Also, it was not possible to discern tension and compression in Figure 8A as indicated in the text and the illustration to the right of the panels. Something appears to be missing here.For a better visualization of tension and compression hotspots in Figure 8A, we now provide the stress ellipses representation on a white background with black cell contours.3) It is essential that the manuscript's results be placed in the context of prior studies. The authors should cite and discuss a paper by Ozawa 2018, which explicitly examines the contribution of myosin IIA to junctional stabilization in MDCK cells.

The Ozawa paper has been now reported and discussed in the Introduction for the major role played by NMIIA in intercellular junction formation (Introduction, last paragraph), as well for the distribution of the two isoforms (subsection “NMIIB preferentially localizes to a junctional actin pool distinct from perijunctional NMIIA-associated contractile fibres”).

In addition, the work by Smutny et al. (which is cited) shows that the formation of cadherin clusters on surfaces requires myosin IIA, not myosin IIB, in apparent contradiction with the present study. The authors should acknowledge and discuss this puzzling difference.

Indeed, Smutny et al. did not notice an effect of NMIIB silencing on cadherin clustering in MCF7 cells spread on E-cadherin in an assay very similar to the one we used here (Figure 6—figure supplement 2). However, our analysis is much more complete with the analysis of cell spreading, actin organisation and traction force measurement, which have may allowed us to detect more subtle effects. Concerning NMIIA silencing, comparing closely Figure 4B in Smutny and our data Figure 6—figure supplement 2A, we can notice a similar alteration of macrocluster formation (cadherin adhesions) and a milder effect on smaller cadherin puncta in both studies. Thus, these apparent divergences are restricted to NMIIB silencing. This may be driven by changes in levels of homo/heterodimerization of NMIIA and NMIIB as discussed now in the subsection “NMIIA is required for the generation of forces at E-cadherin adhesions while NMIIB favours their transmission through F-actin anchoring”.

More broadly, Smutny et al. conclude that "[…] myosin IIB supports integrity of the apical actin ring to prevent fragmentation of the ZA, perhaps by reinforcing it to resist disruptive orthogonal forces." This presages a major outcome of the present study, and should be acknowledged.

We somehow diverge from Smutny et al. interpretations in that we clearly show here that NMIIA is a structural element that assembles actin fiber bundles parallel to the junction that will likely give rise to the apical actin ring observed by Smutny in more polarised cells. We see an effect of NMIIB silencing on the coupling of this actin ring to adhesion complex that may lead to its observed disorganization (Figure 5A, Figure 7) but not its disappearance. Thus NMIIB-mediated coupling may indeed reinforce junctions and prevent them from disruptive forces. This is consistent with our data of TFM on cells seeded non E-cadherin (Figure 6—figure supplement 2). This prediction on the outcome of the present study is now acknowledged (Introduction, last paragraph).

Finally, Gomez et al. examine the specific localization of myosin IIA and IIB in the context of epithelial cell-cell junctions in MCF7 cells (Gomez et al., 2015), and do not appear to observe the differential localization that the authors here report. This difference should likewise be acknowledged.

We do not agree with the reviewers, looking closely at the Figure 1A of Gomez paper (Gomez et al., 2015, one can see localizations of NMIIA and NMIIB imaged by SIM that are fully consistent with Figures 3 and 4 of our manuscript: NMIIB junctional and NMIIA perijunctional. This is also consistent with Figures 1B and 1E of the Ozawa paper.

Overall these differences, which could reflect the use of different cell lines, strongly imply that more caution should be employed in interpreting the present study

Since the differential localization of NMIIA and NMIIB we report here is consistent with the two most recent reports in MCF7 (Gomez) and MDCK (Ozawa) cells and is also observed in Caco2 cells, we do not agree that this differential localization of the two isoforms is restricted to MDCK cells (Figure 3—figure supplement 2B). Moreover, our study focuses on early steps of junction formation, which differs from the 3 other studies performed in polarized cells. This could explain some discrepancies observed here.

4) The authors claim that "NMIIB supports junctional branched actin organization" (Figure 5) is problematic in several ways. First of all, they provide no evidence that the cortical actin network associated with NMIIB is an arp2/3-dependent branched actin network. It is true that other publications showed arp2/3 localizes to junctions, but so do formins. The cortex is probably nucleated by a combination of nucleators, so the authors shouldn't refer to this juxtamembrane network as branched without any evidence.

We modulated our statement (subsection “NMIIB preferentially localizes to a junctional actin pool distinct from perijunctional NMIIA-associated contractile fibres”). Our data allow to say “NMIIB junctional staining colocalizing with β-catenin was associated with a 200 nm to 1 µm thick fuzzy F-actin network (Figure 4A, B), that also contained both Arp2/3 and cortactin (Figure 5B, C), two known molecular markers of branched actin meshwork.” See also changes in text in the Results section and Figure 5 title.

Second, in Figure 5A or elsewhere I don't see any evidence of a disappearing actin network between the control and NMIIB KD images. So, while it is true that NMIIB associates with membrane proximal actin there are no experimental data that support this claim. This claim needs to be discussed more carefully or dropped.

Indeed, we have not been clear enough in the first version of our manuscript. The juxtamembrane actin networks does not disappear in the absence of NMIIB, it is disorganized, basically enlarged as shown in Figure 5B-F and Figure 5—figure supplement 1A. We have modified the text to make it clear now (subsection “NMIIA regulates the organization of perijunctional actin bundles while NMIIB regulates the organization of a juxtamembrane actin layer”).

5) Figure 6 shows that single NMIIA KD cells barely generate any traction forces on ECM. Figure 8 shows the same with cell doublets. It isn't clear what is learned about the role of NMIIA in generating tension at cell-cell junctions from this experiment.

By comparing single cells and doublets, we can conclude that the absence of traction force on ECM on doublets can be explained by the absence of traction force of each cell on the ECM in the case of doublets. This goes with the decreased FA and stress fibres observed both in NMIIA KD single cells and doublets. In addition, these cells could not apply traction forces on E-cadherin substrates. Thus in NMIIA KD doublets, the decrease in intercellular stress can be explained by the impossibility of each cell to apply forces on the substratum as well as on cell-cell contacts due to a lack of organisation of the major contractile apparatus.

In contrast, Figure 6 shows that the traction forces of NMIIB KD cells on ECM are the same as control.

This is indeed the case, in agreement with the reported absence of effect of NMIIB silencing on cell spreading and traction on the ECM (Betapudi, Licate and Egelhoff, 2006; Cai et al., 2006; Jorrisch, Shih and Yamada 2013).

In doublets, Figure 8 shows that the overall traction forces are nearly the same as control, but they observe forces underneath the cell-cell junctions – something not observed in control. This suggests that the force transmission across the cell-cell junction is perturbed and that is why forces are felt at the cell-ECM interface. This should be considered as an explanation.

We agree with the reviewer that the redistribution of forces may be explained by increasing cell-ECM traction forces underneath cell-cell junctions in NMIIB KD cells. Traction forces exerted at the free edges of each cell is then balanced by shared forces transmitted at cell-substrate and cell-cell interfaces in NMIIB KD cells. We added this possible explanation in the revised version of the manuscript.

[Editors' note: further revisions were requested prior to acceptance, as described below.]

The manuscript has been improved but there are some remaining issues that need to be addressed before acceptance, as outlined below:The reviewers agree that you have addressed most of the points raised in the original submission. However, there are still concerns regarding overstated claims of novelty and perhaps interpretations of published data. While you have been careful in the Discussion to emphasize that you are looking at early stages of junction formation, which is an important and new aspect of your work, some of the data concerns junctions in fully polarized cells, which do overlap the work of Yap and Ozawa. Thus, before this manuscript can be accepted, several points need to be addressed in order to place your findings in the proper context of previous papers. None of these points require new experiments, but rather small modifications to the text. Please consider these carefully and modify accordingly.1) The conclusion of the Introduction makes it sounds like this topic has not been investigated previously. A simple fix would be to change "questioned the unexplored functions […]" to "further explore the functions […]" or something similar.

Done: here is the corrected sentence: “Here we further explore the functions of NMII isoforms in epithelial AJ biogenesis using an in vitro system based on chemically-switchable micro-patterns, whereby we can control the time and location of a new contact forming between two single cells on a matrix-coated surface.”

2) In the subsection “NMIIA and NMIIB orchestrate junction biogenesis”, you need to add a sentence acknowledging that Smutny 2010 reported similar findings in mature junctions. Likewise, the phrase "explored the possibility" is misleading given that Smutny had found NMII isoform differences in regulating junctional actin, and this should be acknowledged.

Done, here is the corrected sentence: “NMIIA favours temporal stability whereas NMIIB ensures the straightness and spatial stability of the junctions, which is in agreement with different contributions of NMIIA and NMIIB in mature junctions [Smutny et al., 2010].”

and: “Based on these observations and previous studies [Smutny et al., 2010; Efimova and Svitkina, 2018], we subsequently explored the possibility that NMIIB and NMIIA could differentially regulate actin organization at the junction, thereby maintaining its structural integrity.”

3) There are some differences in the differential localization of the NMII isoforms with the Smutny 2010 and Gomez 2015 papers. It appears that some of this is due to your careful examination of NMIIb colocalized with cadherin, whereas the other papers seem to have looked only at actin. It would be helpful to consider this difference.

Done, here is the added sentence: “Our careful examination by SIM during junction biogenesis revealed precise patterns of NMII, actin and E-cadherin localization whereas other studies mostly focused on actin and NMII in mature junctions [Smutny et al., 2010; Gomez et al., 2015].”

4) At first glance your results with α-catenin would seem to contradict those of the Ozawa 2018 paper. However, since you report a lowering of total α-catenin at junctions (subsection “NMIIA regulates the organization of perijunctional actin bundles while NMIIB regulates the organization of a juxtamembrane actin layer”, last paragraph), there is likely no real discrepancy. However, you should discuss this in the Discussion.

Done, here is the added sentence: “However, we observed a reduced junctional recruitment of α-catenin in NMIIA KD cells, suggesting also a contribution of NMIIA in α-catenin activation in agreement with a previous report [Ozawa, 2018].”